# The representation theory of seam algebras

**Alexis Langlois-Rémillard**[1][⋆][§] **and Yvan Saint-Aubin**[2][†]

**1** Department of Applied Mathematics, Computer Science and Statistics,
Faculty of Sciences, Ghent University, Krijgslaan 281-S9, 9000 Gent, Belgium.
**2** Département de mathématiques et de statistique,
Université de Montréal, Québec, Canada, H3C 3J7.

⋆ Alexis.LangloisRemillard@UGent.be, † yvan.saint-aubin@umontreal.ca

## Abstract

The boundary seam algebras $b_{n,k}(\beta = q + q^{-1})$ were introduced by Morin-Duchesne, Ridout and Rasmussen to formulate algebraically a large class of boundary conditions for two-dimensional statistical loop models. The representation theory of these algebras $b_{n,k}(\beta = q + q^{-1})$ is given: their irreducible, standard (cellular) and principal modules are constructed and their structure explicited in terms of their composition factors and of non-split short exact sequences. The dimensions of the irreducible modules and of the radicals of standard ones are also given. The methods proposed here might be applicable to a large family of algebras, for example to those introduced recently by Flores and Peltola, and Crampé and Poulain d'Andecy.

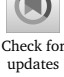
## Contents

§This work was completed as ALR was a student at the Département de mathématiques et de statistique of Université de Montréal.

# 1   Introduction

This article describes the basic representation theory of the family of boundary seam algebras $b_{n,k}(\beta = q + q^{-1})$, for $n \geq 1$, $k \geq 2$ and $q \in \mathbb{C}^{\times}$: their irreducible, standard (cellular) and principal modules are constructed and their structure explicited in terms of their composition factors and of short exact sequences.

The boundary seam algebras, or seam algebras for short, were introduced by Morin-Duchesne, Ridout and Rasmussen [1]. One of their goals was to cast, in an algebraic setting, various boundary conditions of two-dimensional statistical loop models discovered earlier in a heuristic way (see for example [2]). Not only did the authors define diagrammatically the seam algebras, give them a presentation through generators and relations and prove equivalence between the definitions, but they also introduced standard modules over $b_{n,k}$ and computed the Gram determinant of an invariant bilinear form on these modules. All these tools will be used here. Their paper went on with numerical computation of the spectra of the loop transfer matrices under these various boundary conditions. It indicated a potentially rich representation theory.

In its simplest formulation, the seam algebra $b_{n,k}$ is the subset of the Temperley-Lieb algebra $\mathsf{TL}_{n+k}(\beta)$ [3] obtained by left- and right-multiplying all its elements by a Wenzl-Jones projector [4, 5] acting on $k$ of the $n + k$ points. Even though this formulation appears first in [1], the need for some algebraic structure of this type was stressed before by Jacobsen and Saleur [6]. The main goal of their paper was also the study of various boundary conditions for loop models. In a short section at the end of their paper, these authors observed that the blob algebra (see below) can be realized by adding "ghost" strings to link diagrams (their cabling construction) and "tying" them with the first "real" string with a projector. But their goal did not require a formal definition of a new algebra. The definition of the seam algebra $b_{n,k}$ will be given in section 2 and it will be seen there that it is actually a quotient of the blob algebra.

So the seam algebras are yet another variation of the original Temperley-Lieb family. The representation theory of various Temperley-Lieb families has been studied, displaying remarkable richness and diversity: the blob algebra [7, 8], the affine algebra [9], the Motzkin algebra [10], the dilute family [11], etc. Of course it is interesting to see what the representation theory of the seam family hides. And, even though this is a sufficiently intriguing question to justify the present work, there is yet another justification.

In recent years, new families of Temperley-Lieb algebras have been introduced, some having a similar definition to the seam family: they are obtained by left- and right-multiplication of a $\mathsf{TL}_N$, for some $N$, by non-overlapping Wenzl-Jones projectors $P_{i_1}, P_{i_2}, \ldots, P_{i_k}$ with $\sum_{1 \leq j \leq k} i_j = N$. In a sense the seam algebras are the simplest examples of these new families. Two examples of the latter will underline their diverse applications: *(i)* the valence algebras were introduced by Flores and Peltola [12] to characterize monodromy invariant correlation functions in certain conformal field theories and *(ii)* another family of Temperley-Lieb algebras was defined by Crampé and Poulain d'Andecy [13] to understand the centralisers of tensor representations of classical and quantum $sl(N)$. The present paper goes beyond de-

scribing the basic representation theory of the seam algebras: it outlines a method that might help study the representation theory of several other families of algebras.

The main results are stated in section 2, which also gives the definitions of the Temperley-Lieb algebras, the Wenzl-Jones projectors and, of course, the boundary seam algebras. Section 3 is on cellular algebras, a family introduced by Graham and Lehrer [14] to which the seam algebras belong, as will be shown. The proofs there are given so that their generalization to other families like those mentioned above should be straightforward. Section 4 is devoted to the representation theory of the $b_{n,k}(\beta = q + q^{-1})$ when $q$ is a root of unity. This is the difficult case. Section 5 concludes the paper by outlining the key steps of the method in view of applications to other families.

## 2 Definitions and main results

The boundary seam algebras provide examples of algebras obtained from the Temperley-Lieb algebras by left- and right-multiplication by an idempotent. It is natural to put in parallel the basic definitions of $\mathsf{TL}_n$ (section 2.1) and $b_{n,k}$ (section 2.2) and their representation theory (section 2.3 for $\mathsf{TL}_n$ and 2.4 for $b_{n,k}$). This last section states the main results that will be proved in sections 3 and 4.

### 2.1 The family of Temperley-Lieb algebras

The most appropriate definition of the Temperley-Lieb algebras $\mathsf{TL}_n(\beta)$, for the purpose at hand, is its diagrammatic one. An $(n,d)$-diagram is defined as a diagram drawn within a rectangle with $n$ marked points on its left side and $d$ on its right one, these $n+d$ points being connected pairwise by non-crossing links. Two $(n,d)$-diagrams differing only by an isotopy are identified. The set of formal $\mathbb{C}$-linear combinations of $(n,n)$-diagrams will be denoted $\mathsf{TL}_n$. A composition of an $(n,d)$-diagram with a $(m,e)$-diagram is defined whenever $d = m$. Then it is the $(n,e)$-diagram obtained by concatenation and removal of closed loops created by the identification of the middle $d = m$ points, each loop being replaced by an overall factor $\beta \in \mathbb{C}$. (See [15] for examples.) The vector space $\mathsf{TL}_n$ together with this composition is the Temperley-Lieb algebra $\mathsf{TL}_n(\beta)$. For each $n \in \mathbb{N}_{>0}$ and $\beta \in \mathbb{C}$, $\mathsf{TL}_n(\beta)$ is an associative unital $\mathbb{C}$-algebra with the identity diagram Id shown below. It can be proved that $\mathsf{TL}_n(\beta)$, as an algebra, is generated by the identity Id and the following diagrams $E_i$, $1 \leq i < n$:

$$
\mathrm{Id} \;=\; \vcenter{\hbox{\includegraphics{id}}} \qquad \text{and} \qquad E_i \;=\; \vcenter{\hbox{\includegraphics{ei}}}\;.
$$

The dimension of $\mathsf{TL}_n(\beta)$ is equal to the Catalan number $C_n = \frac{1}{n+1}\binom{2n}{n}$. The parameter $\beta$ is often written as $\beta = q + q^{-1}$ where $q \in \mathbb{C}^\times$. Here are the $C_4 = 14$ diagrams spanning $\mathsf{TL}_4$.

$$
\mathrm{Id} \;=\; \vcenter{\hbox{\includegraphics{id4}}}\;,
$$

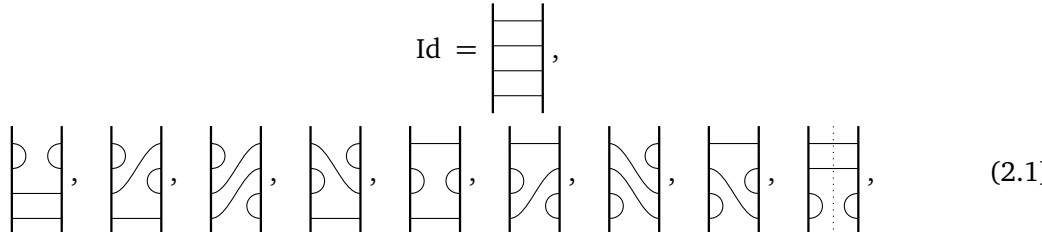

(2.1)



The diagrams have been gathered so that the first line presents the only diagram with 4 links crossing from left to right, the second those that have 2 such links, and the third those that have none. These crossing links are also called through lines or defects.

An elementary observation on concatenation will be crucial: the number of links in an $(n,d)$-diagram that cross from left to right cannot increase upon concatenation with any other diagram. More precisely, if an $(n,d)$-diagram contains $k$ such crossing links and a $(d,m)$-diagram contains $l$ ones, then their concatenation has at most $\min(k,l)$ such links. An $(n,d)$-diagram that has $d$ crossing links is said to be *monic* (and then $n \geq d$) and an $(n,d)$-diagram with $n$ crossing links is called *epic* (and then $n \leq d$). The concatenation of a monic $(n,d)$-diagram with an epic $(d,n)$-one is an $(n,n)$-diagram with precisely $d$ crossing links. The last diagram of the second line above has a dotted vertical line in the middle. It stresses the fact that all diagrams of this second line are concatenations of a monic $(4,2)$-diagram with an epic $(2,4)$-diagram. Similarly, the diagrams of the bottom line are concatenations of a $(4,0)$-diagram with a $(0,4)$-diagram. The single diagram of the top line can also be seen as the concatenation of two epic and monic $(4,4)$-diagrams, that is twice the diagram Id.

The Wenzl-Jones projector $P_n$ [4,5] is an element of $\mathsf{TL}_n(\beta)$ that will play a crucial role in the definition of the seam algebras. (It is also known as the $q$-symmetrizer in other applications [16].) It is constructed recursively as

$$P_1 = \mathrm{Id}, \qquad P_k = P_{k-1} - \frac{[k-1]_q}{[k]_q} P_{k-1} E_{n-k+1} P_{k-1}, \quad \text{for } 1 < k \leq n, \qquad (2.2)$$

where the $q$-numbers $[m]_q = (q^m - q^{-m})/(q - q^{-1})$ were used. Note that this recursive definition of $P_n$ fails whenever $[k]_q = 0$ for some $2 \leq k \leq n$, that is, whenever $q$ is an $2\ell$-root of unity for some $\ell \leq n$. The Wenzl-Jones projector, when it exists, has remarkable properties.

**Proposition 2.1** ([16]). *For $\beta = q + q^{-1}$ with $q$ not a $2\ell$-root of unity for any $\ell \leq n$, the Wenzl-Jones projector $P_n \in \mathsf{TL}_n(\beta)$ is the unique non-zero element of $\mathsf{TL}_n(\beta)$ such that*

$$P_n{}^2 = P_n \qquad and \qquad P_n E_j = E_j P_n = 0, \quad for\ all\ 1 \leq j < n.$$

In fact for $1 \leq k \leq n$, the $P_k$ used to define $P_n$ share some of these properties. For this reason, they will also be referred to as Wenzl-Jones projectors. The properties they share are listed without proof. (See [1,16] and references therein.) First, like $P_n$, the Wenzl-Jones projector $P_k$ is an idempotent. Second, $P_k$ acts trivially on the $n-k$ top links. More precisely, $P_k$ is a linear combination of $(n,n)$-diagrams, each of which has a through line between the first sites on its left and right sides, a through line between the second sites, all the way to a through line between the $(n-k)$-th sites. Thus $P_k$ commutes with the generators $E_i$ for $i \leq n-k-1$. Third, $P_k$ annihilates the $E_i$ with $i \geq n-k+1$. These properties are summed up as

$$\begin{aligned}
&P_k{}^2 = P_k, \\
&P_k E_i = E_i P_k, \qquad \text{when } i \leq n-k-1, \\
&P_k E_i = E_i P_k = 0, \qquad \text{when } i \geq n-k+1.
\end{aligned} \qquad (2.3)$$

Finally the following identities will also be used:

$$E_{n-k} P_k E_{n-k} = \frac{[k+1]_q}{[k]_q} E_{n-k} P_{k-1};$$

$$P_k = \frac{1}{[k]_q}\left([k]_q P_{k-1} - [k-1]_q P_{k-1}E_{n-k+1} + [k-2]_q P_{k-1}E_{n-k+1}E_{n-k+2} + \cdots \right.$$
$$\left. \cdots + (-1)^{k-1}[1]_q P_{k-1}E_{n-k+1}E_{n-k+2}\dots E_n\right).$$

The projector $P_k$ will be represented diagrammatically by

$$P_k := \begin{array}{c}\vdots\\ \end{array} \begin{array}{l}1\\ n-k\\ n-k+1\\ n\end{array} \quad , \quad \text{and thus, for example } P_2 = \quad = \quad - \frac{1}{[2]_q}\quad.$$

With this notation, the last identity reads

$$\begin{array}{c}1\\ \\ k\end{array} = \frac{1}{[k]_q}\left([k]_q \quad - [k-1]_q \quad + [k-2]_q \quad + \cdots + (-1)^{k-1}[1]_q \quad\right). \qquad (2.4)$$

## 2.2 The family of boundary seam algebras

The definition of the *boundary seam algebras* $b_{n,k}(\beta)$ uses the above definitions of $\mathsf{TL}_n$ and of the Wenzl-Jones projectors. It is the subset of $\mathsf{TL}_{n+k}(\beta)$

$$b_{n,k}(\beta) = \langle \mathrm{id}, e_j \mid 1 \le j \le n\rangle \subset \mathsf{TL}_{n+k}(\beta),$$

where $\mathrm{id} = P_k \in \mathsf{TL}_{n+k}$ and $e_j = P_k E_j P_k$ for $1 \le j < n$ and $e_n = [k]_q P_k E_n P_k$[¶]. (The content of the present section follows that of [1].) Clearly this subset is closed under addition and multiplication. It is thus an associative unital algebra, with id as its identity, but it is not a subalgebra of $\mathsf{TL}_{n+k}$ since the identities Id and id of these two algebras are not the same. The $k$ bottom points on both left and right sides of elements of $b_{n,k}$ are called *boundary points* and the other, *bulk points*. (The choice of word comes from the physical interest for boundary seam algebras and will not concern us.) Due to the fact that $P_k$ might not be defined, the range of the two integers $n, k$ and of the complex number $q$ (with $\beta = q + q^{-1}$) will be restricted as follows:

> *(i)* $n \ge 1$, $k \ge 2$ and
>
> *(ii)* $q$ is not a root of unity or, if it is and $\ell$ is the smallest positive integer $\qquad$ (2.5)
> such that $q^{2\ell} = 1$, then $\ell > k$.

Putting $n = 0$ leads to the one-dimensional ideal $\mathbb{C}P_k \subset \mathsf{TL}_k$, and the cases $k = 0$ or 1 correspond to the Temperley-Lieb algebras $\mathsf{TL}_n$ and $\mathsf{TL}_{n+1}$ respectively. With these conditions, the definition is equivalent to left- and right-multiplication by $P_k$, namely: $b_{n,k} \simeq P_k \mathsf{TL}_{n+k} P_k$. The dimension of $b_{n,k}$ is

$$\dim b_{n,k} = \binom{2n}{n} - \binom{2n}{n-k-1}.$$

Both $\mathsf{TL}_n$ and $b_{n,k}$ can be defined through generators and relations. For $\mathsf{TL}_n(\beta)$, $\beta \in \mathbb{C}$, the generators are Id and $E_i$, $1 \le i < n$, with relations

$$\mathrm{Id}\, E_i = E_i\, \mathrm{Id},$$
$$E_i^2 = \beta E_i, \qquad\qquad E_i E_j = E_j E_i, \quad |i-j| > 1, \qquad (2.6)$$
$$E_i E_{i+1} E_i = E_i, \qquad\qquad E_i E_{i-1} E_i = E_i,$$

---

[¶]To ease readability, we shall use capital letters for generators and elements of $\mathsf{TL}_{n+k}$ and small ones for those of $b_{n,k}$.

as long as the indices $i$, $i-1$, $i+1$, and $j$ are in $\{1, 2, \ldots, n-1\}$. The generators for $\mathsf{b}_{n,k}$ are id and $e_i$, $1 \leq i \leq n$, with relations

$$\text{id } e_i = e_i \text{ id},$$

$$
\begin{aligned}
e_i^2 &= \beta e_i, && i < n, & e_i e_j &= e_j e_i, && |i-j| > 1, \\
e_i e_{i+1} e_i &= e_i, && i < n-1, & e_i e_{i-1} e_i &= e_i, && i \leq n-1, \\
e_n^2 &= [k+1]_q e_n, && & e_{n-1} e_n e_{n-1} &= [k]_q e_{n-1}.
\end{aligned}
\tag{2.7}
$$

for indices belonging to $\{1, 2, \ldots, n\}$, and the following relation when $n > k$:

$$\left( \prod_{j=0}^{k} e_{n-j} \right) y_k = \sum_{i=0}^{k-1} (-1)^i [k-i]_q \left( \prod_{j=i+2}^{k} e_{n-j} \right) y_k, \tag{2.8}$$

where the $y_t$ are given recursively by

$$y_0 = [k]_q \text{ id}, \qquad y_1 = e_n,$$

$$[k-t]_q (-1)^t y_{t+1} = \left( \prod_{j=0}^{t} e_{n-j} \right) y_t + \sum_{i=0}^{t-1} (-1)[k-i]_q \left( \prod_{j=i+2}^{t} e_{n-j} \right) y_t. \tag{2.9}$$

Isomorphisms between the two presentations (diagrammatic, and through generators and relations) are explicited in [15] for $\mathsf{TL}_n$ and in [1] for $\mathsf{b}_{n,k}$. For the family of $\mathsf{b}_{n,k}$'s, the defining relations (2.7) and (2.8) allow one to enlarge the domain of the parameters (2.5). However, due to isomorphisms between some pairs $\mathsf{b}_{n,k}$ and $\mathsf{b}_{n,k'}$, the domains (2.5) cover almost all boundary seam algebras defined through generators and relations. Only the family $\mathsf{b}_{n,m\ell}(\beta = q + q^{-1})$, where $\ell$ is the smallest positive integer such that $q^{2\ell} = 1$ and $m$ is a positive integer, is missing. It was shown in [1] that the study of this family can be reduced to that of $\mathsf{b}_{n,\ell}(\beta)$. Little is known about these algebras and it is not clear that the method proposed here applies to them.

The fact that the relations (2.7) are the defining relations for the blob algebra was Jacobsen's and Saleur's key observation in [6] that we alluded to in our introduction. It allowed them, amongst other things, to conjecture in [17] Gram determinant formulas for the blob algebra that were later proved by Dubail [18]. But the algebra $\mathsf{b}_{n,k}(\beta)$ is *not* the blob algebra. Indeed the use of the elements id $= P_k$, $e_j = P_k E_j P_k$ for $1 \leq j < n$ and $e_n = [k]_q P_k E_n P_k$ of $\mathsf{TL}_{n+k}(\beta)$ to define $\mathsf{b}_{n,k}(\beta)$ adds a new relation that is not satisfied by the elements of the corresponding blob algebra. As noted in [1], the discrepancy is seen most easily in the case $k = 1$. Indeed the generators of $\mathsf{b}_{n,1} \simeq \mathsf{TL}_{n+1}$ satisfy the additional relation $e_n e_{n-1} e_n = e_n$, but those of the blob algebra *do not*. When $n > k > 1$, this simple additional relation is replaced by the more complicated (2.8). Thus $\mathsf{b}_{n,k}(\beta)$ is the quotient of the blob algebra by the ideal generated by this relation. Again, as noted in the introduction, Jacobsen and Saleur did not need a formal definition of the algebra generated by the above $e_j$'s. Such a need appeared in [1] where the precise relationship between blob and seam algebras was then made explicit.

## 2.3 The representation theory of the Temperley-Lieb algebras

The representation theory of $\mathsf{TL}_n$ was constructed using three different approaches by Goodman and Wenzl [19], Martin [20], and Graham and Lehrer [9]. Later Ridout and Saint-Aubin [15] gave a self-contained presentation of these results, partially inspired by Graham's and Lehrer's approach and results by Westbury [21]. Here are the main statements.

Let $n \in \mathbb{N}$ and $q \in \mathbb{C}^{\times}$ be fixed. The *standard* or *cellular modules* $\mathsf{S}_n^d$ are modules over the algebra $\mathsf{TL}_n(\beta = q + q^{-1})$. Such modules are defined for each $d$ in the set

$$\Delta_n = \{d \in \mathbb{N} \mid 0 \leq d \leq n \text{ and } d \equiv n \bmod 2\}. \tag{2.10}$$

The module $S_n^d$ is the $\mathbb{C}$-linear span of monic $(n, d)$-diagrams. The action of an $(n, n)$-diagram in $\mathsf{TL}_n(\beta)$ on a monic $(n, d)$-diagram is the composition of diagrams described in section 2.1 (with each closed loop replaced by factor of $\beta$) with the following additional rule: if the $(n, d)$-diagram obtained from the concatenation is not monic, the result is set to zero. Their dimension is

$$\dim S_n^d = \binom{n}{(n-d)/2} - \binom{n}{(n-d-2)/2}.$$

Each of these cellular modules $S_n^d$ carries an invariant bilinear form $\langle \cdot, \cdot \rangle = \langle \cdot, \cdot \rangle_n^d$ defined on $(n, d)$-diagrams and extended bilinearly. For a pair $(v, w)$ of monic $(n, d)$-diagrams, the composition $v^*w$ is first drawn. Here $v^*$ stands for the reflection of $v$ through a vertical mirror. It is thus a $(d, n)$-diagram, and $v^*w$ is a $(d, d)$-diagram. If it is monic, it is equal to $\beta^p$ Id, for some integer $p$, and $\langle v, w \rangle$ is defined to be $\beta^p$. If it is non-monic, then $\langle v, w \rangle = 0$. This bilinear form is symmetric and *invariant* in the sense that, for any $A \in \mathsf{TL}_n$, then $\langle v, Aw \rangle = \langle A^*v, w \rangle$ where, again, $A^*$ is the left-right reflection of $A$. This bilinear form can be identically zero. For $\mathsf{TL}_n(\beta)$, this occurs only when $n$ is even, $d = 0$ and $\beta = 0$. Thus the set

$$\Delta_n^0 = \{d \in \Delta_n \mid \langle \cdot, \cdot \rangle_n^d \not\equiv 0\} \subset \Delta_n \tag{2.11}$$

is identical to $\Delta_n$ unless $n$ is even and $\beta = 0$. A (non-zero) invariant bilinear form carries representation-theoretic information because its radical

$$R_n^d = \{v \in S_n^d \mid \langle v, w \rangle_n^d = 0 \text{ for all } w \in S_n^d\}$$

is a submodule. For the Temperley-Lieb algebras, it gives even more information.

**Proposition 2.2** ( [9]). *The radical $R_n^d$ of the non-zero bilinear form $\langle \cdot, \cdot \rangle_n^d$ is the Jacobson radical of $S_n^d$, that is the intersection of its maximal submodules, and $I_n^d := S_n^d/R_n^d$ is irreducible.*

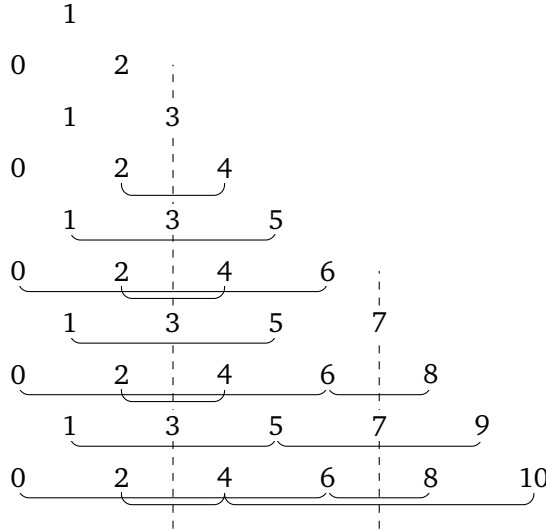

Figure 1: Bratteli diagram for $\mathsf{TL}_n$ for $\ell = 4$ with $n = 1$ as the top line.

One way to identify whether $R_n^d$ is zero or not is to compute the determinant of the *Gram matrix* $\mathcal{G}_n^d$ of $\langle \cdot, \cdot \rangle_n^d$, that is the matrix representing $\langle \cdot, \cdot \rangle_n^d$, say in the basis of monic $(n, d)$-diagrams. This determinant is

$$\det \mathcal{G}_n^d = \prod_{j=1}^{(n-d)/2} \left( \frac{[d+j+1]_q}{[j]_q} \right)^{\dim S_n^{d+2j}}.$$

Clearly $\mathsf{R}_n^d$ might be non-trivial only when $[d+j+1]_q$ is zero for some $j$, namely, when $q$ is some root of unity. This observation is important and justifies the introduction of some vocabulary.

The set $\Delta_n = \{d \in \mathbb{N} \mid 0 \leq d \leq n \text{ and } d \equiv n \bmod 2\}$ is partitioned as follows. If $q$ is not a root of unity, each element of $\Delta_n$ forms its own class in this partition. Suppose that $q$ is a root of unity and let $\ell$ be the smallest positive integer $\ell$ such that $q^{2\ell} = 1$. The letter $\ell$ will be reserved for this integer throughout. If $d \in \Delta_n$ is such that $d+1 \equiv 0 \bmod \ell$, and thus $[d+1]_q = 0$, then $d$ is said to be *critical* and it forms its own class $[d]$ in the partition of $\Delta_n$. If $d$ is not critical, the class $[d]$ consists of images of $d$ in $\Delta_n$ generated by reflections through mirrors positioned at critical integers. In other words, $[d]$ is the *orbit of $d$* under the group generated by these reflections. This information is represented visually in a Bratteli diagram in figure 1 for $\ell = 4$. Each line of the Bratteli diagram contains the elements of the set $\Delta_n$, starting with $\Delta_1$ on the top line. The vertical dashed lines on the diagram go through the critical $d$'s. Elements of the classes for non-critical $d$'s are joined by curly brackets. For $\ell = 4$, the partition of $\Delta_9$ is $\{3\} \cup \{7\} \cup \{1,5,9\}$ and that of $\Delta_{10}$ is $\{0,6,8\} \cup \{2,4,10\}$. Finally, for $d$ a non-critical element of $\Delta_n$, its immediate left and right neighbors in $[d]$ are denoted respectively by $d^-$ and $d^+$. These neighbors might not exist.

The parameter $q$ will be called *generic* (for $\mathsf{TL}_n(\beta = q + q^{-1})$) if it is not a root of unity or, if it is, when all orbits $[d]$ are singletons. An example of the latter case occurs for $\mathsf{TL}_3$ when $\ell = 4$ as can be seen in the Bratteli diagram: on the third line there are no curly brackets and the partition of $\Delta_3$ is $\{\{1\},\{3\}\}$. Note that the condition of genericity can be restated as: $q$ is not a root of unity or $\ell > n$. The latter formulation is usually the one used in the description of the $\mathsf{TL}_n$, but it will turn out that the former will be the one appropriate for the seam algebras. When $q$ is not generic, it will be referred to (somewhat abusively) as being a root of unity and, in this case, it will be understood that the second condition (all orbits $[d]$ are singletons) does not hold.

The following theorem is extracted from the foundational papers [9,19,20]. The projective cover of the irreducible $\mathsf{I}_n^d$ will be denoted by $\mathsf{P}_n^d$.

**Theorem 2.3.** *(a) If $q$ is generic, then $\mathsf{TL}_n(\beta = q + q^{-1})$ is semisimple, the cellular modules are irreducible and the set $\{\mathsf{S}_n^d = \mathsf{I}_n^d \mid d \in \Delta_n^0\}$ forms a complete set of non-isomorphic irreducible modules.*
*(b) If $q$ is a root of unity (with $n \geq \ell$), then $\mathsf{TL}_n(\beta = q + q^{-1})$ is not semisimple. The set $\{\mathsf{I}_n^d = \mathsf{S}_n^d / \mathsf{R}_n^d \mid d \in \Delta_n^0\}$ forms a complete set of non-isomorphic irreducible modules. If $d \in \Delta_n^0$ is critical, then $\mathsf{S}_n^d = \mathsf{I}_n^d = \mathsf{P}_n^d$. If $d$ is not critical, then the two short sequences*

$$0 \longrightarrow \mathsf{I}_n^{d^+} \longrightarrow \mathsf{S}_n^d \longrightarrow \mathsf{I}_n^d \longrightarrow 0$$
$$0 \longrightarrow \mathsf{S}_n^{d^-} \longrightarrow \mathsf{P}_n^d \longrightarrow \mathsf{S}_n^d \longrightarrow 0$$

*are exact and non-split. If $d^+$ is not in $\Delta_n$, then $\mathsf{I}_n^{d^+}$ is set to zero in the first sequence and, if $d^-$ is not in $\Delta_n$, then $\mathsf{S}_n^{d^-}$ is set to zero in the second. As indicated by the first exact sequence, if $\mathsf{R}_n^d$ is not zero, then it is isomorphic to $\mathsf{I}_n^{d^+}$.*

## 2.4 The representation theory of the seam algebras

The representation theory of the boundary seam algebras $\mathsf{b}_{n,k}$ was launched in [1] by constructing the analogues of the cellular modules of the Temperley-Lieb algebras. The cellular modules $\mathsf{S}_{n,k}^d$ over $\mathsf{b}_{n,k}(\beta)$ are spanned by the set $\mathfrak{B}_{n,k}^d$ of non-zero elements of $\{P_k w \mid w \text{ a monic } (n+k,d)\text{-diagram}\}$. Because of the second relation in (2.3), any $P_k w$ with a monic $(n+k,d)$-diagram $w$ that has a link between the boundary points, that is the bottom $k$

points, is zero. So the dimension of the $S_{n,k}^d$, also found in [1], is smaller than that of $S_{n+k}^d$:

$$\dim S_{n,k}^d = \binom{n}{(n+k-d)/2} - \binom{n}{(n-k-d-2)/2} \le \dim S_{n+k}^d.$$

Morin-Duchesne *et al.* also defined a bilinear form $\langle \cdot, \cdot \rangle_{n,k}^d : S_{n,k}^d \times S_{n,k}^d \to \mathbb{C}$. It mimics the definition of the bilinear form on $S_n^d$ defined in the previous section. The bilinear pairing $\langle P_k v, P_k w \rangle_{n,k}^d$ is the factor in front of the monic $(d,d)$-diagram in the concatenation $(P_k v)^*(P_k w) = v^* P_k w$. Like the bilinear form on $S_n^d$, it is symmetric and invariant in the same sense. Morin-Duchesne *et al.* succeeded in computing the determinant of the Gram matrix in the basis $\mathfrak{B}_{n,k}^d$.

**Proposition 2.4** (Prop. D.4, [1]). *The determinant of Gram matrix of the bilinear form* $\langle \cdot, \cdot \rangle_{n,k}^d$ *in the basis* $\mathfrak{B}_{n,k}^d$ *is*

$$\det \mathcal{G}_{n,k}^d = \prod_{j=1}^{\lfloor k/2 \rfloor} \left( \frac{[j]_q}{[k-j+1]_q} \right)^{\dim S_{n,k-2j}^d} \prod_{j=1}^{\frac{n+k-d}{2}} \left( \frac{[d+j+1]_q}{[j]_q} \right)^{\dim S_{n,k}^{d+2j}}. \tag{2.12}$$

The result of this *tour de force* will be useful in what follows. As before, the radical of the bilinear form is defined as

$$R_{n,k}^d = \{ v \in S_{n,k}^d \mid \langle v, w \rangle_{n,k}^d = 0 \text{ for all } w \in S_{n,k}^d \}.$$

Here are the main results of the present paper. Let

$$\Delta_{n,k} = \{ d \in \mathbb{N} \mid 0 \le d \le n+k, \, d \equiv n+k \bmod 2 \text{ and } n+d \ge k \} \tag{2.13}$$

and

$$\Delta_{n,k}^0 = \{ d \in \Delta_{n,k} \mid \text{the bilinear form } \langle \cdot, \cdot \rangle_{n,k}^d \text{ is not identically zero} \}. \tag{2.14}$$

The set $\Delta_{n,k}$ is partitioned exactly as $\Delta_{n+k}$ is. If $q$ is not a root of unity, every element $d$ of $\Delta_{n,k}$ is alone in its class $[d] = \{d\}$. Let $q$ be a root of unity and $\ell$ the smallest positive integer such that $q^{2\ell} = 1$. If $d$ is such that $[d+1]_q = 0$, then $d$ is called critical and $d$ is alone in its class $[d]$. Otherwise the classes $[d]$ are the *orbits* of $d$ under the group generated by reflections through mirrors positioned at critical integers. The $n$-th line in the Bratteli diagram of figure 2 presents the elements in $b_{n,k=8}$ when $\ell = 4$. The points in the shadowed region fail to satisfy the inequality $n+d \ge k$ and are thus excluded from $\Delta_{n,k}$. Elements of a given orbit $[d]$ are joined pairwise by curly brackets. The partitions are easily readable from the diagram. For example the partition of $\Delta_{6,8}$ at $\ell = 4$ is $\{\{2,4,10,12\},\{6,8,14\}\}$.

As for the Temperley-Lieb algebras, the parameter $q$ is called generic if the partition of $\Delta_{n,k}$ contains only singletons. If $q$ is not a root of unity, this is automatically true. If it is not, the Bratteli diagram may be used to quickly construct possible non-trivial orbits and decide whether $q$ is generic. If $q$ is not generic, then it will be referred to as being a root of unity and will not include the cases when the partition of $\Delta_{n,k}$ only contains singletons.

With this definition of genericity, the representation theory of the family of seam algebras $b_{n,k}$ mimics perfectly that of the Temperley-Lieb algebra.

**Theorem 2.5.** *(a) If $q$ is generic, then $b_{n,k}(\beta = q + q^{-1})$ is semisimple, the cellular modules are irreducible and the set $\{ S_{n,k}^d = I_{n,k}^d \mid d \in \Delta_{n,k}^0 \}$ forms a complete set of non-isomorphic irreducible modules.*
*(b) If $q$ is a root of unity (and the partition of $\Delta_{n,k}$ contains at least one orbit $[d]$ of more than one element), then $b_{n,k}(\beta = q + q^{-1})$ is not semisimple. The set $\{ I_{n,k}^d = S_{n,k}^d / R_{n,k}^d \mid d \in \Delta_{n,k}^0 \}$ forms a*

*complete set of non-isomorphic irreducible modules. If $d \in \Delta_{n,k}^0$ is critical, then $\mathsf{S}_{n,k}^d = \mathsf{I}_{n,k}^d = \mathsf{P}_{n,k}^d$. If $d$ is not critical, then the two short sequences*

$$0 \longrightarrow \mathsf{I}_{n,k}^{d^+} \longrightarrow \mathsf{S}_{n,k}^d \longrightarrow \mathsf{I}_{n,k}^d \longrightarrow 0$$

$$0 \longrightarrow \mathsf{S}_{n,k}^{d^-} \longrightarrow \mathsf{P}_{n,k}^d \longrightarrow \mathsf{S}_{n,k}^d \longrightarrow 0$$

*are exact and non-split. If $d^+$ is not in $\Delta_{n,k}$, then $\mathsf{I}_{n,k}^{d^+}$ is set to zero in the first sequence and, if $d^-$ is not in $\Delta_{n,k}$, then $\mathsf{S}_{n,k}^{d^-}$ is set to zero in the second. If $\mathsf{R}_{n,k}^d$ is not zero, then it is isomorphic to $\mathsf{I}_{n,k}^{d^+}$.*

This theorem will be proved over the next two sections.

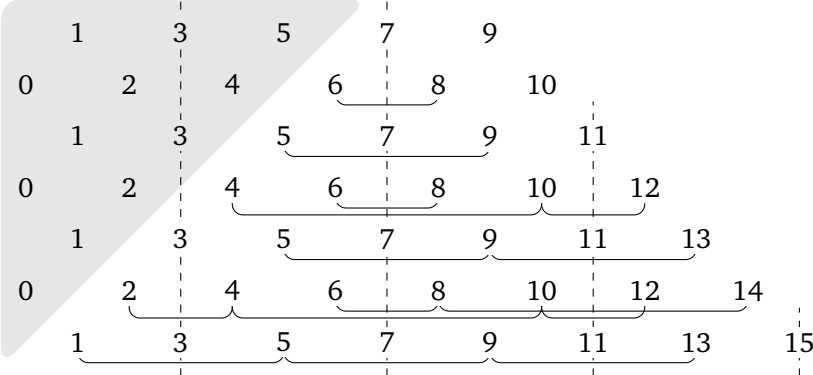

Figure 2: Bratteli diagram for $\mathsf{b}_{n,8}$ with $\ell = 4$ with $n = 1$ as the top line.

# 3 Cellularity of $\mathsf{b}_{n,k}$

One way to prove theorem 2.5 is to reveal the cellular structure of the Temperley-Lieb algebras. Cellular algebras were defined by Graham and Lehrer [14] in part to better understand the bases defined by Kazhdan and Lusztig for the Hecke algebras [22]. Many families of algebras have now been proved to be cellular. The goal of this section is to show that the algebras $\mathsf{b}_{n,k}(q+q^{-1})$ are cellular and to identify the values of $q \in \mathbb{C}^\times$ at which they fail to be semisimple.

## 3.1 The cellular data for $\mathsf{TL}_n$

The definition of cellular algebras is best understood on an example. We recall this definition and give the *cellular datum* for the Temperley-Lieb algebra as example.

**Definition 3.1** (Graham and Lehrer, [14])**.** *Let $R$ be a commutative associative unitary ring. An $R$-algebra $\mathcal{A}$ is called* cellular *if it admits a cellular datum $(\Delta, M, C, *)$ consisting of the following:*

(i) *a finite partially-ordered set $\Delta$ and, for each $d \in \Delta$, a finite set $M(d)$;*

(ii) *an injective map $C : \bigsqcup_{d \in \Delta} M(d) \times M(d) \to \mathcal{A}$ whose image is an $R$-basis of $\mathcal{A}$, with the notation $C^d(s,t)$ for the image under $C$ of the pair $(s,t) \in M(d) \times M(d)$;*

(iii) *an anti-involution $* : \mathcal{A} \to \mathcal{A}$ such that*

$$C^d(s,t)^* = C^d(t,s), \qquad for\ all\ s,t \in M(d); \tag{3.1}$$

(iv) *if $d \in \Delta$ and $s, t \in M(d)$, then for any $a \in \mathcal{A}$,*

$$a C^d(s, t) \equiv \sum_{s' \in M(d)} r_a(s', s) C^d(s', t) \bmod \mathcal{A}^{<d}, \tag{3.2}$$

*where $\mathcal{A}^{<d} = \langle C^e(p, q) \mid e < d; p, q \in M(e) \rangle_R$ and $r_a(s', s) \in R$ is independent of $t$.*

The involution $*$, together with (3.2), yields the equation:

$$C^d(t, s) a^* \equiv \sum_{s' \in M(d)} r_a(s', s) C^d(t, s') \bmod \mathcal{A}^{<d}, \qquad \text{for all } s, t \in M(d) \text{ and } a \in \mathcal{A}. \tag{3.3}$$

Here is the cellular datum for the Temperley-Lieb algebra $\mathsf{TL}_n(\beta)$. Throughout, the commutative ring $R$ will be taken to be the complex field $\mathbb{C}$. Let $\Delta_n$ be the (totally-)ordered set

$$\Delta_n = \begin{cases} \{0 \leq 2 \leq \cdots \leq n-2 \leq n\}, & \text{if } n \text{ is even;} \\ \{1 \leq 3 \leq \cdots \leq n-2 \leq n\}, & \text{if } n \text{ is odd.} \end{cases} \tag{3.4}$$

For $d \in \Delta_n$, the set of monic $(n, d)$-diagrams will be taken as $M(d)$, the anti-involution $*$ is the reflection of diagrams through a vertical mirror, and the map $C : \bigsqcup_{d \in \Delta} M(d) \times M(d) \to \mathsf{TL}_n$ is defined, for $s, t \in M(d)$, to be the $(n, n)$-diagram $C^d(s, t) = st^* \in \mathsf{TL}_n(\beta)$. The basis of $\mathsf{TL}_4$ given in (2.1) shows indeed that the elements of the set $\Delta_4 = \{0, 2, 4\}$ are precisely the number of links crossing from left to right and each line of (2.1) is actually the images by $C$ of $M(4) \times M(4)$, $M(2) \times M(2)$ and $M(0) \times M(0)$, respectively.

The axioms will now be checked. First, the anti-involution respects the equation (3.1) since $(C^d(s, t))^* = (st^*)^* = ts^* = C^d(t, s)$. The application $C$ is injective and surjective on the $\mathbb{C}$-basis of $(n, n)$-diagrams of $\mathsf{TL}_n$. Indeed, the possible number of through lines of any $(n, n)$-diagram lies in $\Delta_n$; an $(n, n)$-diagram with $d$ through lines thus decomposes into a monic $(n, d)$-diagram and an epic $(d, n)$-diagram. For example, the dotted line of the last diagram with $d = 2$ of (2.1) splits this $(4, 4)$-diagram into a monic $(4, 2)$-diagram (the left half) and an epic $(2, 4)$-diagram (the right one).

As observed in section 2.1, concatenation cannot increase the number of the links crossing diagrams. By definition, the subset $\mathsf{TL}_n^{<d}$ is spanned by diagrams with less than $d$ through lines. Thus the axiom (3.2) is trivially verified as it simply reasserts this property of concatenation: the multiplication of any element $A \in \mathsf{TL}_n$ with the element $C^d(s, t)$ with $d$ through lines gives an element with $d$ through lines (that might be zero) plus, maybe, other diagrams in $\mathsf{TL}_n^{<d}$. The coefficient $r_a(s', s)$ in the axiom is then the factor $\beta^m$ coming from the loops closed upon concatenation, and is indeed independent of $t$.

The cellular structure of an algebra $\mathcal{A}$ gives a family of modules, called the cellular modules.

**Definition 3.2.** *Let $\mathcal{A}$ be an $R$-algebra with cellular datum $(\Delta, M, C, *)$ and let $d \in \Delta$. The cellular module $\mathsf{S}^d$ is a free $R$-module with basis $\{v_s \mid s \in M(d)\}$ with $\mathcal{A}$-action given by*

$$a \cdot v_s := \sum_{s' \in M(d)} r_a(s', s) v_{s'}, \quad \text{for all } a \in \mathcal{A}, \tag{3.5}$$

*where $r_a(s', s)$ is the element of $R$ defined in axiom* (3.2).

For the (cellular) Temperley-Lieb algebra $\mathsf{TL}_n(\beta)$, the cellular module $S^d$ just defined coincides with the standard module $\mathsf{S}_n^d$ defined in section 2.3. (This can be checked easily or see [15].)

The coefficients $r_a(s', s)$ defined through axiom (3.2) are used to construct cellular modules, but they are even richer. If $p, s, t$ and $u \in M_{\mathcal{A}}(d)$ for some $d \in \Delta_{\mathcal{A}}$, then equations (3.2) and (3.3) lead to two distinct expressions for the product $C(p,s)C(t,u)$:

$$\sum_{t'} r_{C(p,s)}(t',t)C(t',u) \equiv \sum_{s'} r_{C(u,t)}(s',s)C(p,s') \bmod \mathcal{A}^{<d}. \tag{3.6}$$

Since the image of $C$ is a basis of $\mathcal{A}$, only one term may contribute in each sum, namely the term $t' = p$ in the first and the term $s' = u$ in the second. Thus

$$r_{C(p,s)}(p,t) = r_{C(u,t)}(u,s).$$

Since the left member is independent of $u$, and the right one of $p$, it follows that both of these coefficients depend only on $s$ and $t$. This fact is emphasized by writing

$$C(p,s)C(t,u) \equiv r^d(s,t)C(p,u) \bmod \mathcal{A}^{<d},$$

with $r^d(s,t) := r_{C(p,s)}(p,t) = r_{C(u,t)}(u,s)$.

**Definition 3.3.** *A bilinear form* $\langle \cdot, \cdot \rangle_{\mathcal{A}}^d : S_{\mathcal{A}}^d \times S_{\mathcal{A}}^d \to R$ *on the cellular module* $S_{\mathcal{A}}^d$ *is defined by* $\langle v_s, v_t \rangle = r^d(s,t)$.

This bilinear form plays a central role in the theory of cellular algebra because of the following result.

**Proposition 3.4** (Graham and Lehrer, Prop. 2.4, [14])**.** *The bilinear form* $\langle \cdot, \cdot \rangle_{\mathcal{A}}^d$ *on* $S_{\mathcal{A}}^d$, $d \in \Delta_{\mathcal{A}}$, *has the following properties.*

(i) *It is symmetric:* $\langle x, y \rangle_{\mathcal{A}}^d = \langle y, x \rangle_{\mathcal{A}}^d$ *for all* $x, y \in S_{\mathcal{A}}^d$.

(ii) *It is invariant:* $\langle a^* x, y \rangle_{\mathcal{A}}^d = \langle x, a y \rangle_{\mathcal{A}}^d$ *for all* $x, y \in S_{\mathcal{A}}^d$ *and* $a \in \mathcal{A}$.

(iii) *If* $x \in S^d$ *and* $s, t \in M(d)$, *then* $C^d(s,t)x = \langle v_t, x \rangle_{\mathcal{A}}^d v_s$.

Computing $\langle \cdot, \cdot \rangle_n^d$ on the $TL_n$-module $S_n^d$ is straightforward. The elements $p, s, t$ and $u$ in (3.6) are then elements of $M(d)$, that is, they are monic $(n,d)$-diagrams. In the $(n,n)$-diagram

$$C(p,s)C(u,t) = p(s^* t)u^* \equiv r^d(s,t)C(p,u) \bmod TL_n^{<d},$$

the $(d,d)$-subdiagram $(s^* t)$ may be either

(i) non-monic: then $C(p,s)C(t,u)$ belongs to $TL_n^{<d}$ and $r^d(s,t) = 0$; or

(ii) monic: then $(s^* t)$ is a multiple of the identity $(d,d)$-diagram and the factor is $\beta^m$ where $m$ is the number of loops closed upon concatenation of $s^*$ with $t$. In this case, $r^d(s,t) = \beta^m$.

This bilinear form is precisely the one defined in section 2.3. Proposition 2.2 stated there is in fact a general result that holds for the bilinear form defined in definition 3.3. Let

$$\Delta_{\mathcal{A}}^0 = \{d \in \Delta_{\mathcal{A}} \mid \langle \cdot, \cdot \rangle_{\mathcal{A}}^d \text{ is not identically zero}\}. \tag{3.7}$$

**Proposition 3.5** (Graham and Lehrer, Prop. 3.2, [14])**.** *Let* $\mathcal{A}$ *be a cellular algebra and let* $d \in \Delta_{\mathcal{A}}^0$.
*(a) The radical of the bilinear form* $\langle \cdot, \cdot \rangle_{\mathcal{A}}^d$ *defined by* $R^d = \{x \in S^d \mid \langle x, y \rangle_{\mathcal{A}}^d = 0 \text{ for all } y \in S^d\}$ *is the Jacobson radical of* $S^d$ *and the quotient* $I^d := S^d / R^d$ *is irreducible.*
*(b) The set* $\{I^d \mid d \in \Delta_{\mathcal{A}}^0\}$ *forms a complete set of equivalence classes of irreducible modules of* $\mathcal{A}$.

## 3.2 The cellular data for $b_{n,k}$

The seam algebra $b_{n,k}(\beta = q + q^{-1})$ with parameters as in (2.5) is cellular. In fact it inherits this property and its cellular datum from the cellularity of $\mathsf{TL}_{n+k}(\beta = q + q^{-1})$ described above. That it is cellular follows immediately from the following proposition.

**Proposition 3.6** (König and Xi, Prop. 4.3, [23]). *Let $\mathcal{A}$ be a cellular $R$-algebra with cellular datum $(\Lambda_{\mathcal{A}}, M_{\mathcal{A}}, C_{\mathcal{A}}, *_{\mathcal{A}})$ and $e^2 = e \in \mathcal{A}$ be an idempotent fixed by the involution, that is $e^{*_{\mathcal{A}}} = e$. The algebra $\mathcal{B} = e\mathcal{A}e$ is cellular.*

We provide a proof of their theorem in the special case when the idempotent is the Wenzl-Jones projector $P_k$ and $\mathcal{A} = \mathsf{TL}_{n+k}$ and $\mathcal{B} = b_{n,k}$. Even though it is only a special case of their more general result, the proof displays the cell datum of $b_{n,k}$.

*Proof.* For $d \in \Delta_{\mathcal{A}}$, define the set

$$N(d) = \{s \in M_{\mathcal{A}}(d) \mid P_k C_{\mathcal{A}}^d(s,t) P_k \equiv 0 \bmod \mathcal{A}^{<d} \text{ for all } t \in M_{\mathcal{A}}\} \subset M_{\mathcal{A}}(d).$$

With this definition, the cell datum for $\mathcal{B}$ is as follows. The poset is

$$\Delta_{\mathcal{B}} = \{d \in \Delta_{\mathcal{A}} \mid N(d) \neq M_{\mathcal{A}}(d)\}$$

together with the restriction of partial order on $\Delta_{\mathcal{A}}$. The sets $M_{\mathcal{B}}(d)$ are simply $M_{\mathcal{A}}(d) \setminus N(d)$, for $d \in \Delta_{\mathcal{B}}$. Finally the involution $*_{\mathcal{B}}$ is the restriction of $*_{\mathcal{A}}$ to $\mathcal{B}$ and the map $C_{\mathcal{B}} = \bigsqcup_{d \in \Delta_{\mathcal{B}} \times \Delta_{\mathcal{B}}} M_{\mathcal{B}}(d) \times M_{\mathcal{B}}(d) \to \mathcal{B}$ is

$$C_{\mathcal{B}}^d(s,t) = P_k C_{\mathcal{A}}^d(s,t) P_k.$$

The rest of the proof is devoted to showing that $(\Delta_{\mathcal{B}}, M_{\mathcal{B}}, *_{\mathcal{B}}, C_{\mathcal{B}})$ is a cellular datum for $\mathcal{B}$.

First, $\Delta_{\mathcal{B}}$ is a finite set, and so are the $M_{\mathcal{B}}(d)$'s for all $d \in \Delta_{\mathcal{B}}$. The image of the map $C_{\mathcal{B}}$ is a spanning set for $\mathcal{B}$ since $P_k(\operatorname{im} C_{\mathcal{A}}) P_k$ is. It is also a basis. This rests on the nature of the diagrams that appear in $P_k$. By its recursive construction, $P_k$ has the form $\mathrm{Id} + \sum_i \alpha_i v_i$ where $\alpha_i \in \mathbb{C}$ and $v_i$ are $(n+k, n+k)$-diagrams whose top $n$ sites are joined by the identity diagram with $n$ points. Moreover these $v_i$'s have at least one link tying two boundary left points and another tying two right ones. (Recall that boundary points are the $k$ bottom points of an $(n+k, n+k)$-diagram.)

Let $w$ be an element in the image of $C_{\mathcal{B}}$. It is of the form $P_k C_{\mathcal{A}}(s,t) P_k$ for some $s, t \in M_{\mathcal{B}}(d)$, $d \in \Delta_{\mathcal{B}}$. The diagram $C_{\mathcal{A}}(s,t)$ cannot have any link tying two boundary points, nor one tying right ones, as then $P_k C_{\mathcal{A}} P_k$ would be zero because of (2.3). Thus $w = C_{\mathcal{A}}(s,t) + \sum_i \gamma_i w_i$ with $\gamma_i \in \mathbb{C}$ and where $C_{\mathcal{A}}(s,t)$ has no link between left boundary points and none between right ones, and all the $w_i$'s have such links on either the left or right side, or both. Suppose that the linear combination $\sum_{(s,t)} \alpha_{(s,t)} C_{\mathcal{B}}(s,t)$ vanishes. (The sum over pairs $(s,t)$ may include pairs of different $M_{\mathcal{B}}(d) \times M_{\mathcal{B}}(d)$.) Then, the coefficients of diagrams with no links between boundary points (either on the left side or on the right) must vanish. By the previous observations, this requirement amounts to $\sum_{(s,t)} \alpha_{(s,t)} C_{\mathcal{A}}(s,t) = 0$ which forces $\alpha_{(s,t)} = 0$ for all pairs $(s,t)$, since the image of $C_{\mathcal{A}}$ is a basis of $\mathcal{A}$. The image of $C_{\mathcal{B}}$ is thus a basis of $\mathcal{B}$.

The generators $E_i$ of $\mathsf{TL}_{n+k}$ are clearly invariant under reflection through a vertical mirror. A quick recursive proof shows that $P_k$ is also invariant: $P_k^{*_{\mathcal{A}}} = P_k$. Then

$$C_{\mathcal{B}}(s,t)^{*_{\mathcal{B}}} = (P_k C_{\mathcal{A}}(s,t) P_k)^{*_{\mathcal{B}}} = P_k^{*_{\mathcal{A}}} C_{\mathcal{A}}(s,t)^{*_{\mathcal{A}}} P_k^{*_{\mathcal{A}}} = P_k C_{\mathcal{A}}(t,s) P_k = C_{\mathcal{B}}(t,s).$$

The axiom (3.1) is thus verified for $*_{\mathcal{B}}$.

It remains to check axiom (3.2). Let $b \in \mathcal{B}$. There exists an $A \in \mathcal{A}$ such that $b = P_k A P_k$. For $d \in \Delta_{\mathcal{B}}$ and $s, t \in M_{\mathcal{B}}(d)$, the axiom (3.2) is proven using $P_k^2 = P_k$ and the axiom (3.2) for $\mathcal{A}$:

$$P_k A P_k C_{\mathcal{B}}(s,t) = P_k A P_k^2 C_{\mathcal{A}}(s,t) P_k \overset{(3.2)}{\equiv} P_k \left( \sum_{s' \in M_{\mathcal{A}}(d)} r_{P_k A P_k}(s',s) C_{\mathcal{A}}^d(s',t) \right) P_k \bmod A^{<d}$$

$$\equiv \sum_{s' \in M_{\mathcal{A}}(d)} r_{P_k A P_k}(s',s) P_k C_{\mathcal{A}}^d(s',t) P_k \mod \mathcal{A}^{<d}$$

$$\equiv \sum_{s' \in M_{\mathcal{B}}(d)} r_b(s',s) C_{\mathcal{B}}^d(s',t) \mod \mathcal{B}^{<d},$$

which closes the proof. ∎

**Corollary 3.7.** *The seam algebra* $b_{n,k}(\beta = q + q^{-1})$*, with parameters* $n, k$ *and* $q$ *constrained by* (2.5)*, is cellular.*

The above proof has revealed the cellular datum of $\mathcal{B} = b_{n,k} = P_k \mathsf{TL}_{n+k} P_k \subset \mathcal{A} = \mathsf{TL}_{n+k}$. The set $\Delta_{\mathcal{B}}$ is a subset of $\Delta_{\mathcal{A}}$. It may coincide with or be distinct of $\Delta_{\mathcal{A}}$. For example, $\Delta_{\mathsf{TL}_6} = \{0,2,4,6\}$, but $\Delta_{b_{2,4}} = \{2,4,6\}$, because the set

$$N(0) = \left\{ \text{diagrams} \right\}$$

equals $M_{\mathcal{A}}(0)$ and, thus $0 \notin \Delta_{b_{2,4}}$. Indeed each of these diagrams has two adjacent boundary points tied by a link and the Wenzl-Jones $P_4$ projects each of them to zero. This example shows the way to a simpler characterization of the datum $\Delta_{b_{n,k}}$. For $d \in \Delta_{\mathsf{TL}_{n+k}}$ to be an element of $\Delta_{b_{n,k}}$, there must be a monic $(n+k,d)$-diagram without links between boundary points. The monicity of the diagram takes $d$ points that can all be put at the bottom of the diagram. That way one gets the minimum number $(k-d)$ of boundary points that need to be joined pairwise with some other points. To avoid creating links between boundary points, all of these $(k-d)$ points must be paired with some of the top $n$ points. This is possible if and only if $n \geq k - d$. Thus

$$\Delta_{n,k} = \Delta_{b_{n,k}} = \{d \in \Delta_{\mathsf{TL}_{n+k}} \mid n + d \geq k\}$$
$$= \{d \in \mathbb{N} \mid 0 \leq d \leq n+k, d \equiv n+k \mod 2 \text{ and } n + d \geq k\}. \tag{3.8}$$

From now on, the shorter $\Delta_{n,k}$ will be used instead of $\Delta_{b_{n,k}}$. Similarly $\Delta_n$ will mean $\Delta_{\mathsf{TL}_n}$. These shorter notations match those used in sections 2.3 and 2.4. The points in the shadowed region of the Bratteli diagram in figure 2 are those excluded from the $\Delta_{b_{n,k}}$.

A basis of cellular modules $S_n^d$ over $\mathsf{TL}_n$ was identified to the set of monic $(n,d)$-diagrams. Similar bases for the cellular modules $S_{n,k}^d$ over $b_{n,k}$ are easily identified: a basis for $S_{n,k}^d$ is the subset of monic $(n,d)$-diagrams that have no links between boundary points. These bases can also be identified to (or are in one-to-one correspondence with) the sets $M_{b_{n,k}}(d)$. For $b_{3,2}$, the bases for the three cellular modules $S_{3,2}^5, S_{3,2}^3$ and $S_{3,2}^1$ are respectively:

$$\mathcal{B}_{3,2}^5 = \left\{ \text{diagram} \right\}, \quad \mathcal{B}_{3,2}^3 = \left\{ \text{diagrams} \right\}, \quad \mathcal{B}_{3,2}^1 = \left\{ \text{diagrams} \right\}.$$

As for $\mathsf{TL}_n$, the action of $b_{n,k}$ on $S_{n,k}^d$ obtained formally through definition 3.2 coincides with the composition of diagrams (see section 2.1) with, again, the rule that the the concatenation does not yield $P_k w$ with $w$ a monic $(n+k,d)$-diagram. A few examples are useful. The

first example, the action of $e_1$ on the third vector of $\mathsf{S}^3_{3,2}$, gives zero because monicity is lost through concatenation:

$$\text{(diagram)} \cdot \text{(diagram)} = \text{(diagram)} = \text{(diagram)} = 0.$$

The two other examples are in $\mathsf{S}^1_{3,2}$.

$$\text{(diagram)} \cdot \text{(diagram)} = \text{(diagram)} = \frac{[3]_q}{[2]_q}\text{(diagram)} \quad \text{and} \quad \text{(diagram)} \cdot \text{(diagram)} = \text{(diagram)} = 0.$$

On the left, the closed loop intersected by the projector $P_2$ is removed by using the explicit expression of the Wenzl-Jones projector and, on the right, the result is zero since a link is created tying the two points of the rightmost $P_2$.

Finally definition 3.3 gives the bilinear form $\langle\,\cdot\,,\,\cdot\,\rangle^d_{\mathsf{b}_{n,k}} = \langle\,\cdot\,,\,\cdot\,\rangle^d_{n,k}$. Its expression in the basis $\mathfrak{B}^d_{n,k}$ will be denoted by $\mathcal{G}^d_{n,k}$, and also be called the Gram matrix. With the ordered bases given above, one gets the following matrices:

$$\mathcal{G}^5_{3,2} = (1), \qquad \mathcal{G}^3_{3,2} = \begin{pmatrix} [2]_q & 1 & 0 \\ 1 & [2]_q & 1 \\ 0 & 1 & \frac{[3]_q}{[2]_q} \end{pmatrix}, \qquad \mathcal{G}^1_{3,2} = \begin{pmatrix} [3]_q & \frac{[3]_q}{[2]_q} & \frac{[3]_q}{[2]_q} \\ \frac{[3]_q}{[2]_q} & [3]_q & 0 \\ \frac{[3]_q}{[2]_q} & 0 & [3]_q \end{pmatrix}.$$

The computation of each element requires some practice and we give two examples:

$$\left\langle \text{(diagram)}, \text{(diagram)} \right\rangle^1_{3,2} = \text{(diagram)} = \text{(diagram)} = \text{(diagram)} - \frac{[1]_q}{[2]_q}\text{(diagram)} = \left([2]_q - \frac{1}{[2]_q}\right)\text{(diagram)} = \frac{[3]_q}{[2]_q}\text{(diagram)},$$

where the explicit expression of $P_2$ was used, and

$$\left\langle \text{(diagram)}, \text{(diagram)} \right\rangle^1_{3,2} = \text{(diagram)} = \text{(diagram)} = 0,$$

because of the second relation of (2.3).

## 3.3 The recursive structure of the bilinear form on the cellular $\mathsf{b}_{n,k}$-modules

The goal of this section is to reveal the recursive structure of the Gram matrix $\mathcal{G}^d_{n,k}$ of the bilinear form $\langle\,\cdot\,,\,\cdot\,\rangle^d_{n,k}$ on the cellular $\mathsf{b}_{n,k}$-module $\mathsf{S}^d_{n,k}$. Even though computing the determinant (2.12) of these Gram matrices was an impressive feat and the result will be used below, the recursive form of $\mathcal{G}^d_{n,k}$ given below in lemma 3.8 is one more tool essential to understand the cellular modules. The method is inspired from techniques found in [21] and [15]. The reader might find it worthwhile to have a look at figure 3 below for a graphical interpretation of the change of basis of the following lemma.

**Lemma 3.8.** *When $[d+1]_q \neq 0$ and $d \geq 1$, there exists a unitriangular change of basis matrix $\mathcal{U}$ such that*

$$\mathcal{U}^T \mathcal{G}_{n,k}^d \mathcal{U} = \begin{pmatrix} \mathcal{G}_{n-1,k}^{d-1} & 0 \\ 0 & \frac{[d+2]_q}{[d+1]_q} \mathcal{G}_{n-1,k}^{d+1} \end{pmatrix}. \tag{3.9}$$

*Proof.* The proof is broken into several steps, each being given a short title.

*The new basis.* — The original basis $\mathfrak{B}_{n,k}^d$ is the set of monic $(n+k,d)$-diagrams without link between boundary points and multiplied on the left by $P_k$. It is first partitioned into the set $\mathfrak{F}_1$ of diagrams that have a defect at position 1, and the set $\mathfrak{F}_2$ of diagrams that have a link tying the top point on the left to another below. The new basis $\mathfrak{B}_{n,k}^d{}'$ keeps the set $\mathfrak{F}_1$ unchanged, but replaces the elements of $\mathfrak{F}_2$ by diagrams where the link starting at point 1 and the $d$ defects are acted upon by the projector $P_{d+1}$. These new elements form a set $\mathfrak{F}_2'$ and the ordered basis $\mathfrak{B}_{n,k}^d{}'$ puts the diagrams of $\mathfrak{F}_1$ before those of $\mathfrak{F}_2'$. For example, here is the basis $\mathfrak{B}_{4,2}^2{}'$:

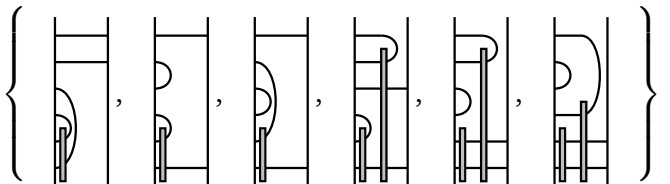

In this example the first three elements form $\mathfrak{F}_1$ and the last three, $\mathfrak{F}_2'$. The added projector $P_{d+1} = P_3$ appears on their right.

*The matrix $\mathcal{U}$ is unitriangular.* — To prove the unitriangularity of $\mathcal{U}$, it is sufficient to show that any element of $\mathfrak{F}_2'$ differs from its corresponding one in $\mathfrak{F}_2$ by an element in $\mathfrak{F}_1$ only. This is done by using the identity (2.4) on the projector $P_{d+1}$ and tracking down what the top link becomes. Here is an example on the first element of $\mathfrak{F}_2'$ of $\mathfrak{B}_{4,2}^2{}'$ (with $d=2$) where the use of (2.4) is confined to the interior of the dotted box:

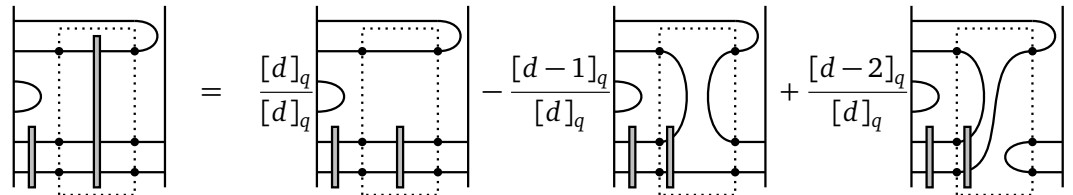

Whichever element of $\mathfrak{F}_2'$ is chosen, the first term of the expansion is the corresponding element in $\mathfrak{F}_2$. This is clearly the case in the above example as the two remaining projectors $P_2$ can be multiplied to give a single $P_2$, because $P_2$ is an idempotent. The general case is more complicated: there will be $d$ defects attached to points above and to the boundary. Relation (2.4) needs to be used repeatedly until only defects attached to the boundary remain so that the remaining projector can be pushed into the $P_k$. All but one of the terms created by repetitive use of (2.4) have a link joining two points to the right; the remaining one is corresponding to the original element of $\mathfrak{F}_2$. The second term of the expansion is always an element of $\mathfrak{F}_1$. Finally all the following terms have only $d-2$ through lines and they are not monic, that is, they are set to zero.

*The sets $\mathfrak{F}_1$ and $\mathfrak{F}_2'$ are mutually orthogonal.* — Any pair $v \in \mathfrak{F}_1$ and $w \in \mathfrak{F}_2'$ is orthogonal. The following diagram, drawn for $\langle v, w \rangle$ with $v$ and $w$ being the third and fourth elements of $\mathfrak{B}_{4,2}^2{}'$, may help follow the argument:

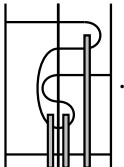

The top through line of $v^*$ enters the projector $P_{d+1}$ from the right. There are thus only $(d-1)$ through lines left in $v^*$ to cross $P_{d+1}$. The two remaining left positions of $P_{d+1}$ must therefore be linked and, by (2.3), $\langle v, w \rangle = 0$.

*The restriction of* $\langle \cdot, \cdot \rangle_{n,k}^d$ *to* $\mathfrak{F}_1$ *is* $\mathcal{G}_{n-1,k}^{d-1}$. — Let $v, w \in \mathfrak{F}_1$ and let $v'$ and $w'$ be the elements of $\mathfrak{B}_{n-1,k}^{d-1}$ obtained from $v$ and $w$ respectively by deleting the top through line. The top through line of $v^*w$ accounts for the monicity of the $(d,d)$-diagram but does not contribute to any factor (it does not close a loop). It can thus be removed and $\langle v, w \rangle_{n,k}^d = \langle v', w' \rangle_{n-1,k}^{d-1}$.

*The restriction of* $\langle \cdot, \cdot \rangle_{n,k}^d$ *to* $\mathfrak{F}_2'$ *is* $\alpha \mathcal{G}_{n-1,k}^{d+1}$ *with* $\alpha = [d+2]_q/[d+1]_q$. — Let $v', w' \in \mathfrak{F}_2'$. The argument will be split according to whether $\langle v', w' \rangle_{n,k}^d$ is zero or not. Recall that when $\langle v', w' \rangle_{n,k}^d$ is non-zero, its value comes from numerical factors that appear through the closing of loops. In the case of $b_{n,k}$, these loops are of two types: those that are intercepted by the projector $P_k$ and those that are not. Each one of the latter type produces a factor $\beta$ and those of the former type are taken care all at once by the identity (equation (D.9) of [1]) that can be shown recursively using (2.3):

$$
\vcenter{\hbox{}} \; n \atop j \atop k-j \quad = \quad \frac{[k+1]_q}{[k-j+1]_q} \; \vcenter{\hbox{}} \; n+j \atop k-j \; . \tag{3.10}
$$

Note that $[k-j+1]_q$ is never zero under the constraints (2.5).

There is a bijection $\psi : \mathfrak{B}_{n-1,k}^{d+1} \to \mathfrak{F}_2'$ obtained as follows: from a diagram in $\mathfrak{B}_{n-1,k}^{d+1}$, a diagram of $\mathfrak{F}_2'$ is given by acting on the right by the Wenzl-Jones projector $P_{d+1}$ and then closing the topmost defect into an arc with a point added at the top of the left side. Let $v, w \in \mathfrak{B}_{n-1,k}^{d+1}$ and let $v' = \psi(v), w' = \psi(w) \in \mathfrak{F}_2'$ be their image under $\psi$. The $(d+1, d+1)$-diagram $v^*w$ may contain closed loops, say $m$ loops that do not intersect $P_k$ and $j$ that do. So

$$
v^*w = \beta^m \frac{[k+1]_q}{[k-j+1]_q} \vcenter{\hbox{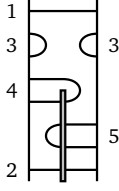}}, \tag{3.11}
$$

where the diagram $D$ on the right-hand side has: $d+1$ points on each of its sides, $d+1$ links joining pairwise these $2(d+1)$ points, a projector $P_{k-j}$ in its middle part, and no loop. The loops that were removed in this exercise also appear in $v'^*w'$ and their removal can be done before or after applying $\psi$ to $v$ and $w$, with the same result. In other words, if $v^*w$ vanishes because the factor $\beta^m[k+1]_q/[k-j+1]_q$ is zero, then so does $v'^*w'$, and vice versa. If this numerical factor is zero, the identity is thus proved.

Assume now that the factor $\beta^m[k+1]_q/[k-j+1]_q$ is not zero. The comparison must then focus on the diagram $D$ on the right-hand side of (3.11) and the diagram $D'$ obtained from it by multiplying both sides by $P_{d+1}$ and closing the top defects coming out of the two $P_{d+1}$. With the removal of closed loops from $v^*w$, the links in $D$ can be deformed to be of one of the five types present in the following diagram: it can contain a through line avoiding $P_{k-j}$, as in (1), or crossing the projector (2); or a link between points on the same side avoiding the projector, as in (3); or crossing it partially (4), or totally (5).

The pairing $\langle v, w\rangle_{n-1,k}^{d+1}$ will be zero if and only if there is at least one link of type (3), (4) or (5), as these are the only ones breaking its monicity. But this statement is also true for $D'$, even after the closing of the top through line: this is immediate for both (3) and (5) because of the second relation in (2.3) (left drawing below) and, for (4), the projector $P_{k-j}$ can be absorbed into $P_{d+1}$ because of the third equation of (2.3) (right drawing):

$$
\text{(diagram)} \quad \text{and} \quad \text{(diagram)} \; .
$$

Therefore the $(d+1, d+1)$-diagram $D$ is monic (and thus leads to a non-zero $\langle v, w\rangle_{n-1,k}^{d+1}$) if and only if $D'$ is monic. It remains to compute the factor between $D$ and $D'$ in the case $D$ is monic. But then, the relations (2.3) and (3.10) for a projector $P_{d+1}$ with one loop give

$$
D = \text{(diagram)} \quad \longrightarrow \quad D' = \text{(diagram)} \; = \; \text{(diagram)} \; = \; \frac{[d+2]_q}{[d+1]_q} \; \text{(diagram)} \, ,
$$

where $[d+1]_q$ is non-zero by hypothesis. The factor $\frac{[d+2]_q}{[d+1]_q}$ is independent of $n$ and $k$. This ends the proof. ∎

Note that, in the previous proof, each step establishing that $\langle v, w\rangle_{n-1,k}^{d+1} = \frac{[d+2]_q}{[d+1]_q}\langle v', w'\rangle_{n,k}^{d}$, $v, w \in \mathfrak{B}_{n-1,k}^{d+1}$ actually proves that $\langle v, w\rangle_{n-1,k}^{d+1}$ and $\langle v', w'\rangle_{n,k}^{d}$ are either both zero or non-zero, except for the last step. Indeed the factor $[d+2]_q$ could be zero.

Here is an example of the factorisation for the Gram matrix $\mathcal{G}_{4,2}^2$. Figure 3 displays the diagrams to be evaluated in the two bases: those on the left are concatenation of elements of the original basis $\mathfrak{B}_{4,2}^2$, those on the right of the new one $\mathfrak{B}_{4,2}^2{}'$. With the use of (3.10), it is easy to evaluate each matrix element. The resulting matrices are respectively

$$
\begin{pmatrix}
[3]_q & \frac{[3]_q}{[2]_q} & 0 & 0 & 0 & 1 \\
\frac{[3]_q}{[2]_q} & [3]_q & \frac{[3]_q}{[2]_q} & \frac{[3]_q}{[2]_q} & 1 & [2]_q \\
0 & \frac{[3]_q}{[2]_q} & [3]_q & 1 & [2]_q & 1 \\
0 & \frac{[3]_q}{[2]_q} & 1 & [3]_q & [2]_q & 1 \\
0 & 1 & [2]_q & [2]_q & [2]_q^2 & [2]_q \\
1 & [2]_q & 1 & 1 & [2]_q & [2]_q^2
\end{pmatrix}
\quad \text{and} \quad
\begin{pmatrix}
[3]_q & \frac{[3]_q}{[2]_q} & 0 & 0 & 0 & 0 \\
\frac{[3]_q}{[2]_q} & [3]_q & \frac{[3]_q}{[2]_q} & 0 & 0 & 0 \\
0 & \frac{[3]_q}{[2]_q} & [3]_q & 0 & 0 & 0 \\
0 & 0 & 0 & \frac{[3]_q}{[2]_q}\frac{[4]_q}{[3]_q} & \frac{[4]_q}{[3]_q} & 0 \\
0 & 0 & 0 & \frac{[4]_q}{[3]_q} & [2]_q\frac{[4]_q}{[3]_q} & \frac{[4]_q}{[3]_q} \\
0 & 0 & 0 & 0 & \frac{[4]_q}{[3]_q} & [2]_q\frac{[4]_q}{[3]_q}
\end{pmatrix} .
$$

The characteristics of the recursive form in lemma 3.8 appear clearly in the second one: the unchanged upper left $3 \times 3$ block, the factor $[d+2]_q/[d+1]_q = [4]_q/[3]_q$ common to all factors in the $3 \times 3$ lower right block and, of course, the two vanishing off-diagonal $3 \times 3$ blocks. The determinants of both matrices agree with that given by formula (2.12) (as they must) and, even though none of their elements contains a factor $[5]_q$, are equal to $[5]_q[4]_q^4/[2]_q^4$.

**Proposition 3.9.** *If $n, k$ are constrained by (2.5) and $d \in \Delta_{n,k}^0$ is such that $[d+1]_q \neq 0$, then*

$$
\det \mathcal{G}_{n,k}^d = \frac{[d+2]_q}{[d+1]_q} \det \mathcal{G}_{n-1,k}^{d-1} \det \mathcal{G}_{n-1,k}^{d+1}. \tag{3.12}
$$

*Proof.* Follows from the unitriangularity of the matrix of change of basis $\mathcal{U}$ and the previous lemma. ∎

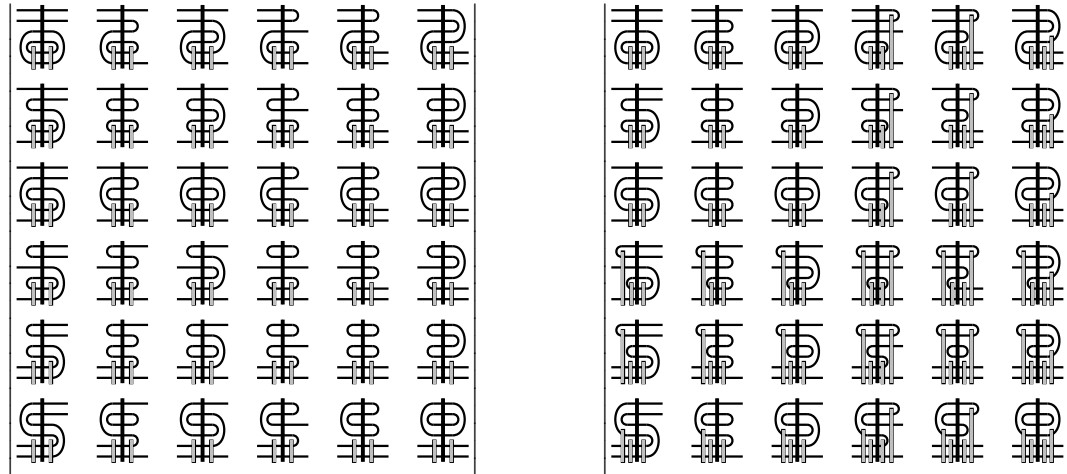

Figure 3: Graphical representation of the determinant of Gram matrix $\mathcal{G}_{4,2}^2$ both in its usual form (left) and after applying the change of basis of lemma 3.8 (right).

With this proposition, it is possible to have a recursive definition of the determinant stated only in terms of seam algebras modules when $[d+1]_q \neq 0$. This result shows that, for a *generic* $q$, the cellular modules $\mathsf{S}_{n,k}^d$ are all irreducible. Indeed it will be checked that the determinant $\det \mathcal{G}_{n,k}^d$ is non-zero for all $d \in \Delta_{n,k}$ (first paragraph of the proof of proposition 4.10). This implies that the radical of $\mathsf{S}_{n,k}^d$ is 0 and thus that $\mathsf{S}_{n,k}^d$ is irreducible for all $d \in \Delta_{n,k}$. The case when $q$ is a root of unity requires more work. The next section is devoted to this problem.

However before closing the present section, two seemingly unrelated questions are answered: when are $\Delta_{n,k}$ and $\Delta_{n,k}^0$ distinct sets? And are the cellular modules $\mathsf{S}_{n,k}^d$ cyclic? Identity (3.10), used to prove the recursive form of $\mathcal{G}_{n,k}^d$, is also key toward the answer of the first question.

**Proposition 3.10.** *Let* $d \in \Delta_{n,k}$. *The bilinear form* $\langle \cdot, \cdot \rangle_{n,k}^d$ *is identically zero if and only if* $d < k$ *and* $k+1 \equiv 0 \bmod \ell$.

*Proof.* For any pair of basis elements $v, w \in \mathfrak{B}_{n,k}^d$, the product $\langle v, w \rangle$ is either zero or of the form $\beta^i \frac{[k+1]_q}{[k-j+1]_q}$ because of the identity (3.10). Here $i$ is the number of closed loops that do not go through the projector $P_k$ and $j$, the number of closed loops that do. Recall that, for the product $\langle v, w \rangle$ to be non-zero, the diagram $v^* w$ needs to be proportional to the $(d, d)$-diagram identity, and then the product is the factor multiplying the identity. Amongst all possible diagrams $v^* w$, there always exists at least one that has $i = 0$. Indeed a pair of a $(d, n+k)$-diagram $v^*$ and an $(n+k, d)$-diagram $w$ can be constructed that has this property. Draw first $\min(d, k-1)$ through lines on the bottom of the diagram (these all go through $P_k$) and if $d \geq k$, the remaining $d-k+1$ at the top of the bulk. (Recall that the bulk are the upper $n$ points, the boundary the lower $k$ points.)

If $k > d$, then this has left untaken the $n$ sites of the bulk and $k-d$ sites of the boundary (below the dashed line in the examples below). Since elements of $\Delta_{n,k}$ satisfy $n+d \geq k$, then the number of untaken boundary sites is smaller than that of the untaken bulk ones. Since $d$ shares the parity of $n+k$, there are an even number of sites left and it is possible to go through them all, bulk and boundary, by drawing $k-d$ loops, each intersecting once $P_k$. (The figure on the left gives an example of the case $k > d$. The diagram on the left of the vertical line is

$v^*$ and on the right $w$.)

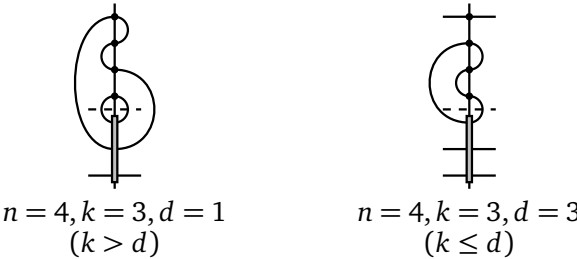

$$n = 4, k = 3, d = 1$$
$$(k > d)$$

$$n = 4, k = 3, d = 3$$
$$(k \leq d)$$

The case $k \leq d$ is split in two. If $d = n + k$, then the module is one-dimensional and the bilinear form non-zero. Assume then that $d \leq n + k - 2$. The drawing of the $d$ defects has thus left untouched $n - (d - k + 1)$ bulk sites (which is an odd positive integer) and 1 boundary site. It is thus possible to draw one loop going through all the remaining sites. (The figure on the right gives an example of the case $k \leq d$.) In these two cases, the product $\langle v, w \rangle$ is equal to $\frac{[k+1]_q}{[k-j+1]_q}$ for some $j$. (In the above examples, $j = 2$ for the case $k > d$ and $j = 1$ for $k \leq d$.) The existence of such a pair $v, w \in \mathfrak{B}^d_{n,k}$ shows that, if $[k+1]_q \neq 0$, then $\langle \cdot, \cdot \rangle^d_{n,k}$ is not identically zero. This statement holds whether or not $\beta = 0$.

The form $\langle \cdot, \cdot \rangle^d_{n,k}$ may then be identically zero only if $[k+1]_q = 0$ which is equivalent to $k + 1 \equiv 0 \bmod \ell$. It will thus be identically zero if and only if *all* diagrams $v^*w$, with $v, w \in \mathfrak{B}^d_{n,k}$, contain a loop going through the projector $P_k$. This situation occurs only when the boundary sites of any diagrams $v^*w$ cannot all be occupied by through lines, that is, if $d < k$. ∎

Note that it is possible for $\beta$ and $[k+1]_q$ to be simultaneously zero. Then $q = \pm i$ and $k = 1$, because $\ell = 2$ muse be greater than $k$. The case $k = 1$ was omitted by (2.5) for this reason; it corresponds to $b_{n,1}(0) = \mathsf{TL}_{n+1}(0)$ and is treated in [15]. Finally the condition $k + 1 \equiv 0 \bmod \ell$ is simply $\ell = k + 1$ due to the constraint *(ii)* in (2.5).

**Proposition 3.11.** *The cellular modules* $\mathsf{S}^d_{n,k}$ *over* $b_{n,k}$ *are cyclic.*

*Proof.* The proof is split according to whether the bilinear form $\langle \cdot, \cdot \rangle^d_{n,k}$ is identically zero or not. Suppose first that it is not zero. Then, for any non-zero element $z$ in $\mathsf{S}^d_{n,k} \setminus \mathsf{R}^d_{n,k}$, there exists a $y \in \mathsf{S}^d_{n,k}$ such that $\langle y, z \rangle^d_{n,k} \neq 0$. In fact, since $b_{n,k}$ is considered as an algebra over $\mathbb{C}$, an element $y$ can be chosen such that $\langle y, z \rangle^d_{n,k} = 1$. Let $x \in \mathsf{S}^d_{n,k}$ be any other element. Then

$$x = x \langle y, z \rangle^d_{n,k} = x(y^*z) = (xy^*)z \in b_{n,k}z.$$

The module $\mathsf{S}^d_{n,k}$ is thus cyclic and any non-zero element in $\mathsf{S}^d_{n,k} \setminus \mathsf{R}^d_{n,k}$, a generator.

Now suppose that $\langle \cdot, \cdot \rangle^d_{n,k}$ is identically zero. The preceding proposition puts the following four constraints on the the three integers $n, k$ and $d$: $0 \leq d \leq n + k$; $k \leq n + d$; $d < k$, and $[k+1]_q = 0$. The proof consists here on showing that the element $z \in \mathfrak{B}^d_{n,k}$ constructed as follows is a generator of $\mathsf{S}^d_{n,k}$. Since $z$ has $d$ defects, it must have $m = (n+k-d)/2$ arcs. These $m$ arcs are drawn as nested arcs joining, if $m \leq k$, the first $m$ boundary and last $m$ bulk sites and, if $m > k$, the bottom $2m$ of the $n + k$ points of $z$. Defects take over the remaining points

of $z$. Here are two examples, one for each case:

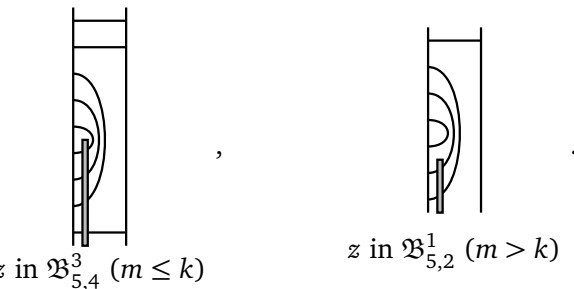

The rest of the proof constructs, for a given $v \in \mathfrak{B}_{n,k}^d$, an element $a_v \in \mathsf{b}_{n,k}$ such that $a_v z = v$. This will establish that $z$ is a generator and thus that $\mathsf{S}_{n,k}^d$ is cyclic. Here are the few steps of this construction. They will be exemplified for $z$ and the following $v$ in $\mathfrak{B}_{13,5}^4$:

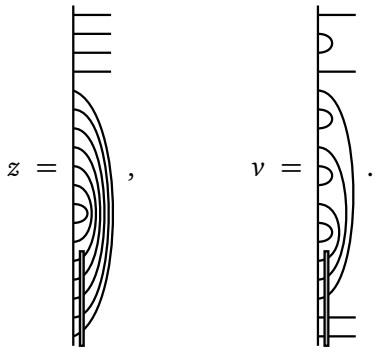

The integers $n = 13$, $k = 5$ and $d = 4$ satisfy the inequalities recalled in the previous paragraph.

For any diagram $v$, the diagram $a_v$ will have its $k$ boundary nodes joined by through lines to those of $z$ so that no closed loop intersecting $P_k$ are created. This can be seen as the *zeroth* step of the construction. The *first* step puts the links in $a_v$ that will give to $a_v z$ the bulk defects of $v$. In the *second* step, if $v$ has more defects in the boundary than $z$, then arcs on the right side of $a_v$ are added to connect the upper defects of $z$ to the associated boundary defects of $v$. (The result of these two first steps are shown in the leftmost diagram below.) Note that the defects of $v$ are now reproduced by the concatenation of $a_v$ and $z$.

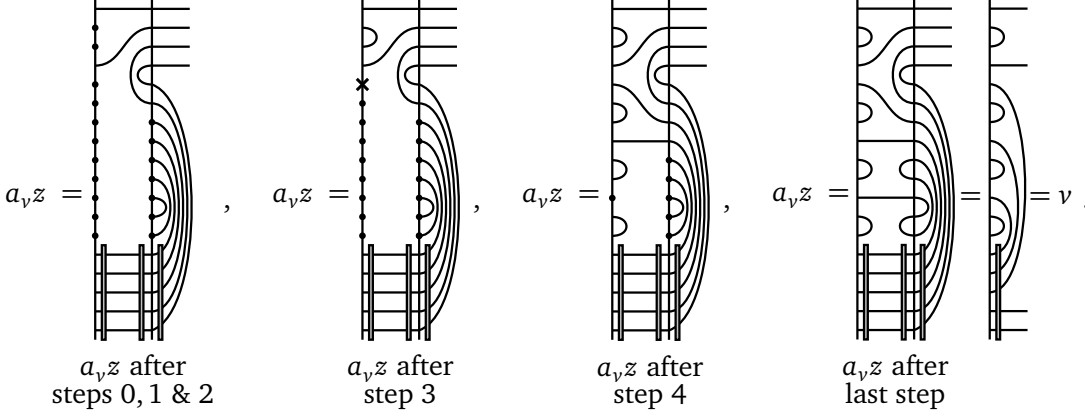

For the next step, find the highest point ✗ reached by an arc in $v$ that goes through the boundary. (We have marked the point in the second diagram above by such a ✗.) All arcs in $v$ above this point ✗ need to be in $a_v z$ and the *third* step simply draws them on the left side of $a_v$. (The result is shown in the second diagram above.) The feature of $v$ that remain to be

reproduced in $a_v z$ are the arcs, either joining two bulk nodes, or one bulk and one boundary nodes, that are not above ✖. A direct check shows that the numbers of free points on either sides of $a_v$ have the same parity. (In the example, they are respectively 9 and 7.) These conditions insure that the *fourth* step can proceed. The arcs of $v$ with both extremities in the bulk are reproduced in the left part of $a_v$. These will not be modified upon concatenation. Then all but one of the remaining sites are joined to the topmost boundary arcs of $z$ still not connected to $a_v$. As they must not intersect, there is only one way to do so. The *fifth* and last step is to draw a curve that starts at the only remaining site on the left of $a_v$ and visits all the remaining points on its right side before reaching its final destination, the highest node amongst the remaining ones of $z$. This is always possible because, when there is a single boundary arc remaining to close, the number of free sites on the right side of $a_v$ is odd and thus a "snake" can be drawn to visit them all. (In the example, this number is 5.) There might be more than one such snaking curve, but there is always at least one because of the nestedness of the remaining arcs in $z$. The resulting $a_v$ of the example is seen in the rightmost diagram. It is now clear that all features of $v$ have been reproduced. The concatenation $a_v z$ has closed no loops and the equality $a_v z = v$ is strict, that is, no factor $\beta^i [k+1]_q / [k-j+1]_q$ (that might have been vanishing) has appeared. Thus $z$ is a generator of $\mathsf{S}_{n,k}^d$. ∎

It will be proved later on that, when $d \notin \Delta_{n,k}^0$, the cellular module $\mathsf{S}_{n,k}^d$ is irreducible. Therefore any $z \in \mathfrak{B}_{n,k}^d$, or any non-zero element of the module for that matter, is a generator. The advantage of the one chosen in the proof is that, for any diagram $v$, the corresponding $a_v$ can be chosen to be a diagram of $\mathsf{b}_{n,k}$, and not a linear combination of diagrams.

# 4 The representation theory of $\mathsf{b}_{n,k}(\beta = q + q^{-1})$ at $q$ a root of unity

In this section, $q$ will be a root of unity and $\ell$ the smallest positive integers such that $q^{2\ell} = 1$. Throughout the parameters $n, k$ and $q$ are constrained by (2.5).

## 4.1 Dimensions of radicals and irreducibles

Lemma 3.8 leads naturally to a recursive formula for the dimensions of the radical $\mathsf{R}_{n,k}^d$ of the cellular modules $\mathsf{S}_{n,k}^d$, and thus of the irreducible quotients $\mathsf{I}_{n,k}^d = \mathsf{S}_{n,k}^d / \mathsf{R}_{n,k}^d$. Of course, lemma 3.8 may be used as long as the constraint $[d+1]_q \neq 0$ is satisfied. The case $[d+1]_q = 0$ must be dealt with separately and, for this goal, formula (2.12) is needed.

**Proposition 4.1.** *When $d$ is critical, that is, when $[d+1]_q = 0$, the cellular module $\mathsf{S}_{n,k}^d$ is irreducible.*

*Proof.* The condition $[d+1]_q = 0$ implies the existence of an $m \in \mathbb{N}$ such that $d+1 = m\ell$. The determinant of the Gram matrix (2.12) is given by two products. As $k < \ell$, the numerators $[j]_q$ of the first product are never 0 since their range is limited to $j \leq \lfloor k/2 \rfloor < \ell$. The numerators in the second product are of the form $[d+j+1]_q$ and, again, an integer $m' \in \mathbb{N}$ such that $j = m'\ell$ should exist for $[d+j+1]_q$ to vanish. However, when such an integer exists, the quotient turns out to be non-zero:

$$\frac{[d+1+j]_q}{[j]_q} = \frac{[(m+m')\ell]_q}{[m'\ell]_q} = \frac{[m+m']_{q^\ell}}{[m']_{q^\ell}} \frac{[\ell]_q}{[\ell]_q},$$

because for any $r \in \mathbb{N}$,

$$[r\ell]_q = \frac{q^{r\ell} - q^{-r\ell}}{q - q^{-1}} = \left( \frac{(q^\ell)^r - (q^\ell)^{-r}}{q - q^{-1}} \right) \left( \frac{q^\ell - q^{-\ell}}{q^\ell - q^{-\ell}} \right) = [r]_{q^\ell} [\ell]_q.$$

As $q^\ell = \pm 1$, the number $[m]_{q^\ell}$ (defined as $\lim_{q^\ell \to \pm 1}[m]_{q^\ell}$) is non-zero. The determinant is thus non-zero and the radical of $S_{n,k}^d$ is trivial. The result then follows from proposition 3.5. ∎

Using lemma 3.8, recursive formulas are obtained for the dimension of the radical.

**Proposition 4.2.** *The dimension of the radical $R_{n,k}^d$ of the cellular module $S_{n,k}^d$, for $d \in \Delta_{n,k}^0$, is given by*

$$\dim R_{n,k}^d = \begin{cases} 0, & \text{if } [d+1]_q = 0; \\ \dim R_{n-1,k}^{d-1} + \dim S_{n-1,k}^{d+1}, & \text{if } [d+1]_q \neq 0 \text{ and } [d+2]_q = 0; \\ \dim R_{n-1,k}^{d-1} + \dim R_{n-1,k}^{d+1}, & \text{if } [d+1]_q \neq 0 \text{ and } [d+2]_q \neq 0. \end{cases} \quad (4.1)$$

*Proof.* The case $[d+1]_q = 0$ has been dealt with in the previous proposition. The dimension of the radical is the dimension of the kernel of the Gram matrix. Lemma 3.8 has put the Gram matrix in block-diagonal form and the dimension of its kernel is the sum of those of the kernels of the two diagonal blocks. If $[d+2]_q \neq 0$, it is simply the sum $\dim R_{n-1,k}^{d-1} + \dim R_{n-1,k}^{d+1}$. If $[d+2]_q = 0$, then the lower block $[d+2]_q/[d+1]_q \mathcal{G}_{n-1,k}^{d+1}$ is zero and the dimension is $\dim R_{n-1,k}^{d-1} + \dim S_{n-1,k}^{d+1}$. ∎

Lemma 3.8 has shown (somewhat implicitly) that $\dim S_{n,k}^d = \dim S_{n-1,k}^{d-1} + \dim S_{n-1,k}^{d+1}$ since these are the sizes of the diagonal blocks appearing in (3.9). Because $I_{n,k}^d = S_{n,k}^d/R_{n,k}^d$, and thus $\dim I_{n,k}^d = \dim S_{n,k}^d - \dim R_{n,k}^d$, the previous proposition has an immediate consequence.

**Corollary 4.3.** *The dimension of the irreducible module $I_{n,k}^d$, for $d \in \Delta_{n,k}^0$, is given by*

$$\dim I_{n,k}^d = \begin{cases} \dim S_{n,k}^d, & \text{if } [d+1]_q = 0; \\ \dim I_{n-1,k}^{d-1}, & \text{if } [d+1]_q \neq 0 \text{ and } [d+2]_q = 0; \\ \dim I_{n-1,k}^{d-1} + \dim I_{n-1,k}^{d+1}, & \text{if } [d+1]_q \neq 0 \text{ and } [d+2]_q \neq 0. \end{cases} \quad (4.2)$$

These results parallel those for the Temperley-Lieb algebras (proposition 5.1 and corollary 5.2 of [15]). They will play a similar role in the characterization of non-trivial submodules of cellular modules when $q$ is a root of unity.

## 4.2 Graham's and Lehrer's morphisms

The previous section has shown that, when $q$ is a root of unity, some cellular modules $S_{n,k}^d$ are reducible. This raises the question of characterizing their structure. In particular, is the radical $R_{n,k}^d$ itself reducible? In their study of the affine Temperley-Lieb algebras, Graham and Lehrer [9] constructed a non-trivial morphisms between cellular $TL_n$-modules at $q$ a root of unity. Because of the relationship between $b_{n,k}$ and $TL_{n+k}$, this family of morphisms will answer the question of the structure of the radical $R_{n,k}^d$.

The morphism of Graham and Lehrer is now recalled. The first definition describes a partial order on the links of a $(t,s)$-diagram.

**Definition 4.4.** *Let $D$ be a $(t,s)$-diagram and $F(D)$ the set of links of $D$. Two elements $x, y \in F(D)$ are ordered $x \leq y$ if $x$ lies in the convex hull of $y$, namely if $y$, as an arc, contains $x$ or if $y$, as a through line, is below $x$.*

The definition is easier to understand through an example. Let $D$ be the $(5,3)$-diagram:

$$D = \begin{array}{c} x \\ y \\ z \end{array} \quad \text{(diagram)} \quad w \quad \longrightarrow \quad \begin{array}{c} z \quad y \quad x \qquad\qquad w \end{array} \text{(diagram)} \, .$$

The diagram on the right was obtained by turning the left side of $D$ clockwise by $90°$ and its right one counterclockwise by the same angle. Here $F(D) = \{x, y, z, w\}$ and an element is smaller or equal to another if the former is contained in the latter. So, for this $D$, the set $F(D)$ is partially ordered by

$$x \leq x, \qquad x \leq y, \qquad x \leq z, \qquad y \leq y, \qquad y \leq z, \qquad z \leq z, \qquad w \leq w.$$

The set $F(D)$ endowed with such a partial order is an example of a forest, that is, if $x \leq y$ and $x \leq z$, then $y \leq z$ or $z \leq y$. The following proposition due to Stanley [24] states a remarkable property of forests.

**Proposition 4.5.** *Let $P$ be a forest of cardinality $n$; for $y \in P$, denote the number of elements of $P$ which are less than or equal to $y$ by $h_y$. The rational function*

$$H_P(q) := \frac{[n]_q!}{\prod_{y \in P} [h_y]_q} \tag{4.3}$$

*is a Laurent polynomial with integer coefficients.*

Recall that $[n]_q$ is the $q$-number $(q^n - q^{-n})/(q - q^{-1})$, and $[n]_q!$, the $q$-factorial $\prod_{1 \leq j \leq n}[j]_q$. With these definitions and results, Graham's and Lehrer's morphism can be described.

**Theorem 4.6** (Graham and Lehrer, Coro. 3.6, [9]). *Let $q \in \mathbb{C}^\times$ be a root of unity, $\ell$ the smallest positive integer such that $q^{2\ell} = 1$, and $\mathsf{TL}_n(\beta)$ be the Temperley-Lieb algebra. If $t, s \in \Lambda_n$ are such that $t < s < t + 2\ell$ and $s + t \equiv -2 \bmod 2\ell$, then there exists a morphism $\theta : \mathsf{S}_n^s \to \mathsf{S}_n^t$ given, for $v \in \mathsf{S}_n^s$, by*

$$\theta(v) = \sum_{\substack{w : s \leftarrow t \\ monic}} \mathrm{sg}(w) H_{F(w)}(q) v w, \tag{4.4}$$

*where $\mathrm{sg}(w)$ is a sign*[‖]. *Moreover $\theta(\mathsf{S}_n^s) = \mathrm{rad}\,\mathsf{S}_n^t$, with the exception that, if $t = 0$ and $\beta = 0$, then $\theta(\mathsf{S}_n^s) = \mathsf{S}_n^t$.*

Note that the condition on the integers $s$ and $t$ is precisely that they be immediate neighbors in an orbit under reflection through vertical critical lines (see section 2.3). Indeed the midpoint between $s$ and $t$ sits at $(s + t)/2 = (-2 + 2m\ell)/2 = m\ell - 1$, for some $m$, and thus on a critical vertical line. In the notation of section 2.3, $s^- = t$ or $t^+ = s$.

The next step is to construct a similar morphism between $\mathsf{b}_{n,k}$-modules. The following result, characterizing the *restriction functor*, a standard tool in the representation theory of associative algebras, will be the key element. (See [25] for a standard treatment of the theory.)

**Proposition 4.7.** *Let $\mathcal{A}$ be an algebra, $e \in \mathcal{A}$ be an idempotent and $\mathcal{B} = e\mathcal{A}e$. The functor $\mathrm{res}_e : \mathrm{mod}\text{-}\mathcal{A} \to \mathrm{mod}\text{-}\mathcal{B}$ that sends a (finitely-generated) module $V$ to $eV$ and a morphism $f : V \to W$ to $\mathrm{res}_e(f) : eV \to eW$ defined by $ev \mapsto ef(v)$ is exact.*

The importance of this functor in the present case is partially revealed by the following result.

**Lemma 4.8.** *The restriction functor $\mathrm{res}_{P_k}$ establishes an isomorphism between the restriction of the radical of the $\mathsf{TL}_{n+k}$-module $\mathsf{S}_{n+k}^d$ and the $\mathsf{b}_{n,k}$-module $\mathsf{R}_{n,k}^d$:*

$$\mathsf{R}_{n,k}^d \simeq \mathrm{res}_{P_k}(\mathsf{R}_{n+k}^d), \qquad \text{for all } d \in \Delta_{n,k}.$$

---

[‖]The sign $\mathrm{sg}(w)$ will not play any role in the following. It is sufficient to know that it can be recovered from the $\mathbb{C}$-algebras isomorphism between $\mathsf{TL}_n(q + q^{-1})$ (used here) and $\mathsf{TL}_n(-q - q^{-1})$ (used by Graham and Lehrer) knowing the sign is always $+1$ in the $-q - q^{-1}$ case.

*Proof.* If the bilinear form $\langle \cdot, \cdot \rangle_{n+k}^d$ on the $\mathsf{TL}_{n+k}$-module $\mathsf{S}_{n+k}^d$ is identically zero, then so is $\langle \cdot, \cdot \rangle_{n,k}^d$ on the $\mathsf{b}_{n,k}$-module $\mathsf{S}_{n,k}^d$. Then $\mathsf{R}_{n,k}^d = \mathsf{S}_{n,k}^d = \mathrm{res}_{P_k} \mathsf{S}_{n+k}^d = \mathrm{res}_{P_k} \mathsf{R}_{n+k}^d$. Suppose then that the bilinear form on $\mathsf{S}_{n+k}^d$ is not identically zero. Let $v \in \mathsf{R}_{n+k}^d$. Then

$$\langle P_k v, P_k w \rangle_{n,k}^d = \langle P_k v, P_k w \rangle_{n+k}^d = \langle v, P_k w \rangle_{n+k}^d = 0$$

because of the invariance property in proposition 3.4 and the fact that the bilinear form on the $\mathsf{b}_{n,k}$-module is the restriction of the one on the $\mathsf{TL}_{n+k}$-module. Hence $\mathsf{R}_{n,k}^d \supset \mathrm{res}_{P_k}(\mathsf{R}_{n+k}^d)$. Now if $P_k v \in \mathsf{R}_{n,k}^d$, then for any $w \in \mathsf{R}_{n+k}^d$

$$\langle P_k v, w \rangle_{n+k}^d = \langle P_k v, P_k w \rangle_{n,k}^d = 0$$

and $\mathsf{R}_{n,k}^d \subset \mathrm{res}_{P_k}(\mathsf{R}_{n+k}^d)$. ∎

If $\mathrm{rad}\, \mathsf{M}$ denotes the Jacobson radical of the module $\mathsf{M}$, then the previous lemma can be reformulated as

$$\mathrm{rad}(\mathrm{res}_{P_k} \mathsf{S}_{n+k}^d) \simeq \mathrm{res}_{P_k}(\mathrm{rad}\, \mathsf{S}_{n+k}^d).$$

The restriction functor of proposition 4.7 carries the morphism $\theta$ of theorem 4.6 into one between $\mathsf{b}_{n,k}$-modules. But is it a non-trivial morphism? This will be the difficult part of the proof ahead.

**Proposition 4.9.** *Let $q \in \mathbb{C}^\times$ be a root of unity, $\ell > 1$ the smallest positive integer such that $q^{2\ell} = 1$, and $\mathsf{b}_{n,k}(\beta)$ be the seam algebra. If $t, s \in \Lambda_{n,k}$ are such that $t < s < t + 2\ell$ and $s + t \equiv -2 \mod 2\ell$, then Graham's and Lehrer's morphism $\theta$ gives rise to a non-trivial morphism $\mathrm{res}_{P_k}(\theta) : \mathsf{S}_{n,k}^s \to \mathsf{S}_{n,k}^t$ given, for $v \in \mathsf{S}_{n,k}^s$, by*

$$\mathrm{res}_{P_k}(\theta)(v) = \sum_{\substack{w:s\leftarrow t \\ monic}} \mathrm{sg}(w) H_{F(w)}(q) P_k v w. \tag{4.5}$$

*Proof.* The restriction functor $\mathrm{res}_{P_k}$ is applied to Graham's and Lehrer's morphism $\theta : \mathsf{S}_{n+k}^s \to \mathsf{S}_{n+k}^t$. Functoriality gives the existence of a morphism between $\mathsf{S}_{n,k}^s$ and $\mathsf{S}_{n,k}^t$ such that the following diagram commutes:

$$
\begin{array}{ccc}
\mathsf{S}_{n+k}^s & \xrightarrow{\;\theta\;} & \mathsf{S}_{n+k}^t \\
{\scriptstyle \mathrm{res}_{P_k}}\downarrow & & \downarrow{\scriptstyle \mathrm{res}_{P_k}} \\
\mathsf{S}_{n,k}^s & \xrightarrow[\mathrm{res}_{P_k}\theta]{} & \mathsf{S}_{n,k}^t
\end{array}
\;.
$$

Hence

$$
\begin{aligned}
\mathsf{R}_{n,k}^t &\simeq \mathrm{res}_{P_k}(\mathsf{R}_{n+k}^t), && \text{by lemma 4.8} \\
&= \mathrm{res}_{P_k}(\mathrm{rad}\, \mathsf{S}_{n+k}^t), && \text{as } \ell > 1 \\
&\simeq \mathrm{res}_{P_k}(\theta(\mathsf{S}_{n+k}^s)), && \text{by theorem 4.6} \\
&\simeq \mathrm{res}_{P_k}(\theta)(\mathrm{res}_{P_k} \mathsf{S}_{n+k}^s), && \text{by functoriality} \\
&= \mathrm{res}_{P_k}(\theta)(\mathsf{S}_{n,k}^s) = \mathrm{im}(\mathrm{res}_{P_k}(\theta))
\end{aligned}
$$

and there is an isomorphism between the image of $\mathrm{res}_{P_k} \theta$ and $\mathsf{R}_{n,k}^t$; it will thus be sufficient to prove that $\dim \mathsf{R}_{n,k}^t$ is not zero to prove that $\mathrm{res}_{P_k}(\theta)$ is non-trivial.

Proposition 4.2 indicates that three cases are to be considered. First $[t+1]_q = 0$, that is, $t$ is critical. This cannot happen as $t$ and $s$ are in a non-critical orbit under reflection. Second $[t+2]_q = 0$ and thus, by (4.1),

$$\dim \mathsf{R}_{n,k}^t = \dim \mathsf{R}_{n-1,k}^{t-1} + \dim \mathsf{S}_{n-1,k}^{t+1}. \tag{4.6}$$

As $t + 1 \leq s - 1 \leq n + k - 1$, the dimension $\dim R_{n,k}^t \geq S_{n-1,k}^{t+1}$ is surely not 0. Third $[t+2]_q \neq 0$. In this last case, equation (4.1) gives

$$\dim R_{n,k}^t = \dim R_{n-1,k}^{t-1} + \dim R_{n-1,k}^{t+1}. \tag{4.7}$$

The upper index of $R_{n-1,k}^{t+1}$ can further increase by $m + 2$ uses of the same relation, such that $[t+m+2]_q = 0$:

$$\dim R_{n,k}^t = \dim R_{n-1,k}^{t-1} + \dim R_{n-2,k}^t + \cdots + \dim R_{n-m-1,k}^{t+m-1} + \dim S_{n-m-1,k}^{t+m+1}. \tag{4.8}$$

(An example of this recursive process follows the proof.) The method to get the term $\dim S_{n-m-1,k}^{t+m+1}$ insures that $t + m + 1$ belongs to $\Delta_{n-m-1,k}$. To see this, note first that the condition $n + d \geq k$ in the definition (3.8) of $\Delta_{n,k}$ remains satisfied at each step. Second, since $[t + m + 2]_q = 0$, the index $t + m + 1$ is critical and $s$ is thus equal to $t + 2(m+1)$. Since $s \in \Delta_{n,k}$, the condition $s \leq n + k$ implies $t + m + 1 \leq n - m - 1 + k$ and $t + m + 1 \in \Delta_{n-m-1,k}$. Hence, $\dim R_{n,k}^t$ is non-zero as $S_{n-m-1,k}^{t+m+1}$ is non-trivial. The non-triviality of $\mathrm{res}_{P_k}(\theta)$ follows. ∎

Figure 4 provides an example of the recursive process used in the proof. It is drawn for $b_{4,3}$ with $\ell = 5$, $s = 7$ and $t = 1$. The morphism $\mathrm{res}_{P_3} \theta : S_{4,3}^7 \to S_{4,3}^1$ is non-trivial as the dimension of $R_{4,3}^1$, isomorphic to its image, is larger or equal to $\dim S_{1,3}^4 = 1$, as can be seen by multiple applications of (4.1):

$$\begin{aligned}
\dim R_{4,3}^1 &= \dim R_{3,3}^0 + \dim R_{3,3}^2 \\
&= \dim R_{3,3}^0 + \dim R_{2,3}^1 + \dim R_{2,3}^3 \\
&= \dim R_{3,3}^0 + \dim R_{2,3}^1 + \dim R_{1,3}^2 + \dim S_{1,3}^4.
\end{aligned}$$

The following Bratteli diagram displays the process with full lines representing applications of proposition 4.2 and circles the indices of the modules (radicals or cellular) appearing in the last line of the above equation. As before, the dashed vertical line is critical.

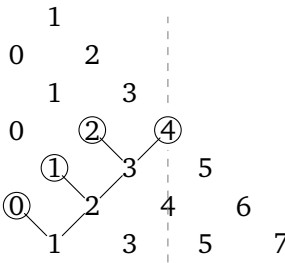

Figure 4: Recursive process for $b_{4,3}$ with $\ell = 5$, $s = 7$ and $t = 1$.

The existence of this family of morphims leads to the structure of the cellular modules over $b_{n,k}$. Proposition 4.1 showed that $S_{n,k}^d$ is irreducible if $d$ is critical. The next proposition studies the case $d$ non-critical.

**Proposition 4.10.** *If $d \in \Delta_{n,k}$ is non-critical, then the short sequence of $b_{n,k}$-modules*

$$0 \longrightarrow I_{n,k}^{d^+} \longrightarrow S_{n,k}^d \longrightarrow I_{n,k}^d \longrightarrow 0 \tag{4.9}$$

*is exact and non-split and thus $R_{n,k}^d \simeq I_{n,k}^{d^+}$. (Note that, if $d \notin \Delta_{n,k}^0$, then $R_{n,k}^d = S_{n,k}^d$ and the module $I_{n,k}^d$ in the above short sequence is understood to be 0.)*

*Proof.* Denote by $[d]$ the orbit of $d$ under the reflection through mirrors at critical integers and by $d_r \in \Delta_{n,k}$ the rightmost (or the largest) integer in this orbit. By definition $d_r^+$ is not in $\Delta_{n,k}$. Let $m \in \mathbb{N}$ be such that $(m-1)\ell - 1 < d_r < m\ell - 1$. The reflection $d_r^+$ of $d_r$ through the critical integer $m\ell - 1$ is determined by $(d_r + d_r^+)/2 = m\ell - 1$. Since $d_r^+$ is not in $\Delta_{n,k}$, then $n + k < d_r^+ = 2m\ell - 2 - d_r$ and thus

$$\frac{n+k+d_r}{2} + 1 < m\ell.$$

The second product of $\det \mathcal{G}_{n,k}^{d_r}$ (see (2.4)) is the only one that can vanish. The numerators in this product run from $[d_r + 2]_q$ to $[(n+k+d_r)/2 + 1]_q$ and thus never vanishes. Hence $\mathsf{S}_{n,k}^{d_r}$ is irreducible. Again, if $d_r^+$ is not in $\Delta_{n,k}$, it is not in $\Delta_{n+k}$ either so that the cellular $\mathsf{TL}_{n+k}$-module $\mathsf{S}_{n+k}^{d_r}$ is also irreducible. In this case, the short exact sequence of theorem 2.3 is simply $0 \to 0 \to \mathsf{S}_{n+k}^{d_r} \to \mathsf{I}_{n+k}^{d_r} \to 0$ and applying the exact restriction functor $\mathrm{res}_{P_k}$ to it gives

$$0 \longrightarrow 0 \longrightarrow \mathsf{S}_{n,k}^{d_r} \longrightarrow \mathrm{res}_{P_k}(\mathsf{I}_{n+k}^{d_r}) \longrightarrow 0,$$

which proves that $\mathrm{res}_{P_k}(\mathsf{I}_{n+k}^{d_r}) \simeq \mathsf{I}_{n,k}^{d_r}$. If $d_r^-$ is not in $\Delta_{n,k}$, then the proof is finished for this orbit.

If the left neighbor $d_r^-$ belongs to $\Delta_{n,k}$, then the short exact sequence for $\mathsf{S}_{n+k}^{d_r^-}$ given in theorem 2.3 is

$$0 \longrightarrow \mathsf{I}_{n+k}^{d_r} \longrightarrow \mathsf{S}_{n+k}^{d_r^-} \longrightarrow \mathsf{I}_{n+k}^{d_r^-} \longrightarrow 0,$$

where $\mathsf{I}_{n+k}^{d_r} \simeq \mathsf{R}_{n+k}^{d_r^-}$, also by theorem 2.3. The restriction functor thus gives

$$0 \longrightarrow \mathrm{res}_{P_k}(\mathsf{I}_{n+k}^{d_r}) \longrightarrow \mathsf{S}_{n,k}^{d_r^-} \longrightarrow \mathrm{res}_{P_k}(\mathsf{I}_{n+k}^{d_r^-}) \longrightarrow 0.$$

It has just been proved that $\mathrm{res}_{P_k}(\mathsf{I}_{n+k}^{d_r}) \simeq \mathsf{I}_{n,k}^{d_r}$ and lemma 4.8 states that $\mathsf{R}_{n,k}^{d_r^-} \simeq \mathrm{res}_{P_k}(\mathsf{R}_{n+k}^{d_r^-}) = \mathrm{res}_{P_k}(\mathsf{I}_{n+k}^{d_r})$, thus showing that $\mathsf{R}_{n,k}^{d_r^-} \simeq \mathsf{I}_{n,k}^{d_r}$. The first isomorphism theorem gives

$$\mathrm{res}_{P_k}(\mathsf{I}_{n+k}^{d_r^-}) \simeq \mathsf{S}_{n,k}^{d_r^-} / \mathrm{res}_{P_k}(\mathsf{I}_{n+k}^{d_r}) \simeq \mathsf{S}_{n,k}^{d_r^-} / \mathsf{R}_{n,k}^{d_r^-} \overset{\mathrm{def}}{=} \mathsf{I}_{n,k}^{d_r^-},$$

thus giving

$$0 \longrightarrow \mathsf{I}_{n,k}^{d_r} \longrightarrow \mathsf{S}_{n,k}^{d_r^-} \longrightarrow \mathsf{I}_{n,k}^{d_r^-} \longrightarrow 0.$$

If $d_r^- \notin \Delta_{n,k}^0$, then $\mathsf{R}_{n,k}^{d_r^-}$ is the whole module $\mathsf{S}_{n,k}^{d_r^-}$ and $\mathsf{I}_{n,k}^{d_r^-}$ should be understood as 0 in the sequence, therefore giving $\mathsf{S}_{n,k}^{d_r^-} \simeq \mathsf{I}_{n,k}^{d_r}$. Otherwise, the proof of the exactness of this sequence for $\mathsf{S}_{n,k}^{d_r^-}$ has given $\mathrm{res}_{P_k}(\mathsf{I}_{n+k}^{d_r^-}) \simeq \mathsf{I}_{n,k}^{d_r^-}$, like the proof for the existence of an exact sequence for $\mathsf{S}_{n,k}^{d_r}$ had given $\mathrm{res}_{P_k}(\mathsf{I}_{n+k}^{d_r}) \simeq \mathsf{I}_{n,k}^{d_r}$. So the present paragraph can be repeated for any element $d$ of the orbit $[d_r]$, thus closing the proof of the existence of the exact sequence (4.9).

It remains to show that these short exact sequences do not split. Proposition 3.11 has shown that the cellular module $\mathsf{S}_{n,k}^d$ is cyclic for all $d \in \Delta_{n,k}$. Suppose then that $z$ is a generator and that $\mathsf{S}_{n,k}^d$ is a direct sum $\mathsf{A} \oplus \mathsf{B}$ of two submodules. The element $z$ can be written as $z_\mathsf{A} + z_\mathsf{B}$ with $z_\mathsf{A} \in \mathsf{A}$ and $z_\mathsf{B} \in \mathsf{B}$. If both $z_\mathsf{A}$ and $z_\mathsf{B}$ were in $\mathsf{R}_{n,k}^d$, then $\mathsf{b}_{n,k} z = \mathsf{b}_{n,k} z_\mathsf{A} \oplus \mathsf{b}_{n,k} z_\mathsf{B} \subset \mathsf{R}_{n,k}^d$, contradicting the fact the $z$ is a generator of $\mathsf{S}_{n,k}^d$. Then, one of $z_\mathsf{A}$ and $z_\mathsf{B}$ is not in $\mathsf{R}_{n,k}^d$ and, by the argument above, is a generator of $\mathsf{S}_{n,k}^d$. If it is $z_\mathsf{A}$, then $\mathsf{B}$ must be zero, and if it is $z_\mathsf{B}$, then it is $\mathsf{A}$ that must be zero. Hence $\mathsf{S}_{n,k}^d$ is indecomposable and the exact sequence (4.9) does not split. ∎

Here is an example. The algebra $b_{4,2}$ with $\ell = 3$ has four cellular modules. Its line in the Bratteli diagram reads:

$$S^0_{4,2} \qquad S^2_{4,2} \qquad S^4_{4,2} \qquad S^6_{4,2} \quad .$$

The modules $S^2_{4,2}$ and $S^6_{4,2}$ are irreducible, the first because the integer 2 is critical, the second because 6 does not have a right neighbor in its orbit.

Because $k + 1 = \ell$, the bilinear form $\langle \cdot, \cdot \rangle^0_{4,2}$ is identically zero. But the "renormalized" Gram matrix is not. For example, at $q = e^{2\pi i/3}$,

$$\lim_{q \to e^{2\pi i/3}} \frac{\mathcal{G}^0_{4,2}}{[k+1]_q} = \begin{pmatrix} -1 & 1 & 0 \\ 1 & -1 & 1 \\ 0 & 1 & -1 \end{pmatrix},$$

whose determinant is 1. This shows that $S^0_{4,2}$ is also irreducible. The pairs $(t,s) = (0,4)$ and $(4,6)$ both satisfy the hypotheses of proposition 4.9 and there are thus two morphisms

$$\theta_1 : S^6_{4,2} \longrightarrow S^4_{4,2}, \qquad\qquad \theta_2 : S^4_{4,2} \longrightarrow S^0_{4,2}.$$

We shall use the following bases of $S^0_{4,2}$, $S^4_{4,2}$ and $S^6_{4,2}$:

$$\mathfrak{B}^0_{4,2} = \left\{ \quad \right\}, \qquad \mathfrak{B}^4_{4,2} = \left\{ \quad \right\} \quad \text{and} \quad \mathfrak{B}^6_{4,2} = \left\{ \quad \right\},$$

as well as the sets of monic $(4,0)$- and $(6,4)$-diagrams:

$$\mathfrak{B}_{4 \leftarrow 0} = \left\{ \quad \right\}, \qquad \mathfrak{B}_{6 \leftarrow 4} = \left\{ \quad \right\}.$$

The action of the morphism $\theta_1$ on the unique element $v$ of the basis $\mathfrak{B}^6_{4,2}$ is given by a sum over the five elements $w_i$ of the set $\mathfrak{B}_{6 \leftarrow 4}$:

$$\theta_1(v) = \sum_{i=1}^{5} \text{sg}(w_i) H_{F(w_i)}(q) v w_i.$$

The coefficients given by proposition 4.5 are

$$H_{F(w_1)}(q) = \frac{[5]_q!}{[1]_q[2]_q[3]_q[4]_q[5]_q} = 1, \qquad H_{F(w_2)}(q) = \frac{[5]_q!}{[1]_q[1]_q[3]_q[4]_q[5]_q} = [2]_q,$$

$$H_{F(w_3)}(q) = \frac{[5]_q!}{[1]_q[2]_q[1]_q[4]_q[5]_q} = [3]_q, \qquad H_{F(w_4)}(q) = \frac{[5]_q!}{[1]_q[2]_q[3]_q[1]_q[5]_q} = [4]_q,$$

$$H_{F(w_5)}(q) = \frac{[5]_q!}{[1]_q[2]_q[3]_q[4]_q[1]_q} = [5]_q.$$

Note that the contribution of $w_5$ will cancel because $P_2 v w_5 = 0$. With the proper signs and replacing the value of the $q$-numbers at $q = e^{\pi i/3}$ ($[2]_q = 1, [3]_q = 0$ and $[4]_q = -1$), the

morphism is

$$\theta_1(v) = \quad - \quad + \quad .$$

The morphism $\theta_2$ is given on the four elements $x_i$ of the basis $\mathfrak{B}_{4,2}^4$ by a sum on the two elements $z_1, z_2$ of the set $\mathfrak{B}_{4 \leftarrow 0}$. As $H_{F(z_1)} = [2]_q$ and $H_{F(z_2)} = 1$, the morphism is defined by

$$\theta_2(v_1) = [2]_q \quad - \quad = - \quad , \qquad \theta_2(v_2) = - \quad ,$$

$$\theta_2(v_3) = - \quad , \qquad \theta_2(v_4) = [2]_q \quad - \quad .$$

Note that the image of $\theta_1$ lies in the kernel of $\theta_2$:

$$\theta_2(\theta_1(v)) = \theta_2(v_1 - v_2 + v_4) = \theta_2(v_1) - \theta_2(v_2) + \theta_2(v_4) = 0.$$

## 4.3 Projective covers

This last section is devoted to the study of projective modules, or more precisely, the indecomposable ones, also called the principals. The theory of cellular algebras over an algebraically closed field like $\mathbb{C}$ provides key information for this task. The standard definitions will be recalled, an example for $b_{4,2}$ with $\ell = 3$ will be worked through, and the main theorem will then be proved.

Let M be a $\mathcal{A}$-module of finite dimension. A filtration of M

$$0 = M_0 \subset M_1 \subset \cdots \subset M_m = M$$

is a *composition series* if each quotient $M_i / M_{i-1}$ is an irreducible module. These quotients are called *composition factors*. The *composition multiplicity* $[M : I]$ of an irreducible $\mathcal{A}$-module $I$ in M is the number of composition factors isomorphic to $I$ in a composition series of M. The Jordan-Hölder theorem assures that it is well-defined.

The regular module $_{\mathcal{A}}A$ is the algebra $\mathcal{A}$ seen as a left module on itself. In its decomposition as a sum of indecomposable modules

$$_{\mathcal{A}}\mathcal{A} \simeq A_1 \oplus \cdots \oplus A_m, \tag{4.10}$$

each summand $A_i$ is called a *principal (indecomposable) modules*. They are projective modules. To each irreducible module $I$, there is one and only one, up to isomorphism, principal module P such that $I \simeq P / \operatorname{rad} P$.

Let $\mathcal{A}$ be a cellular algebra. The composition multiplicities of its cellular and principal modules are intimately related. Define the decomposition matrix of its cellular modules by $\mathbf{D} = \left( \mathbf{D}_{d,e} := [S^d : I^e] \right)_{\{d \in \Delta, e \in \Delta^0\}}$. The order of both indices $d$ and $e$ respects the partial order on $\Delta_{\mathcal{A}}$. Axiom (3.2) constrains the structure of the matrix $\mathbf{D}$.

**Proposition 4.11** (Graham and Lehrer, Prop. 3.6, [14])**.** *The matrix* $\mathbf{D}$ *is upper unitriangular:* $\mathbf{D}_{d,e} = 0$ *if* $d > e$ *and* $\mathbf{D}_{d,d} = 1$.

Indecomposable projective modules on cellular algebras admit a special filtration.

**Lemma 4.12** (Mathas, Lemma 2.19, [26])**.** *Let* $\mathsf{P}$ *be any projective* $\mathcal{A}$-*module and let* $\delta = |\Delta|$. *The module* $\mathsf{P}$ *admits a filtration of* $\mathcal{A}$-*modules*

$$0 = \mathsf{P}_0 \subset \mathsf{P}_1 \subset \mathsf{P}_2 \subset \cdots \subset \mathsf{P}_\delta = \mathsf{P},$$

*in which each quotient* $\mathsf{P}_i/\mathsf{P}_{i-1}$ *is a direct sum of isomorphic copies of a given cellular module* $\mathsf{S}^d$, *with each* $d \in \Lambda$ *appearing once. (Some of these direct sums might be* 0.)

Let $\mathsf{P}^d$ be the principal module associated to $d \in \Delta^0$. It is the projective cover of $\mathsf{I}^d$: $\mathsf{P}^d/\operatorname{rad}\mathsf{P}^d \simeq \mathsf{I}^d$. Consider now the Cartan matrix $\mathbf{C} = \big(\mathbf{C}_{d,e} := [\mathsf{P}^d : \mathsf{I}^e]\big)_{\{d\in\Delta^0, e\in\Delta^0\}}$. Here is the relation between composition multiplicities of cellular and principal modules for the cellular algebra $\mathcal{A}$.

**Theorem 4.13** (Graham and Lehrer, Thm. 3.7, [14])**.** *The matrices* $\mathbf{C}$ *and* $\mathbf{D}$ *are related by* $\mathbf{C} = \mathbf{D}^t\mathbf{D}$.

Examples for $\mathsf{b}_{n,k}$ are given before the general results are proved. When $\mathsf{b}_{n,k}$ is semisimple, then each cellular module is irreducible and the Wedderburn-Artin theorem gives the decomposition of the algebra, as a module over itself:

$$\mathsf{b}_{n,k} \simeq \bigoplus_{d\in\Delta_{n,k}} (\dim \mathsf{S}^d_{n,k})\mathsf{S}^d_{n,k}.$$

Each cellular module is thus a principal module when the algebra is semisimple.

We pursue the example of the algebra $\mathsf{b}_{4,2}$ with $\ell = 3$. The corresponding line in the Bratteli diagram was given in the previous section. It was noted earlier that the three modules $\mathsf{S}^0_{4,2}$, $\mathsf{S}^2_{4,2}$ and $\mathsf{S}^6_{4,2}$ are irreducible. The proposition 4.10 gives the isomorphisms

$$\operatorname{rad}\mathsf{S}^4_{4,2} = \mathsf{R}^4_{4,2} \simeq \mathsf{I}^6_{4,2} = \mathsf{S}^6_{4,2} \qquad \text{and} \qquad \mathsf{S}^0_{4,2} \simeq \mathsf{I}^4_{4,2}.$$

A word of warning: the use of only $\mathsf{S}^0_{4,2}$ instead of $\mathsf{R}^0_{4,2}$ is deliberate, as the bilinear form $\langle\cdot,\cdot\rangle^0_{4,2}$ is identically zero. Thus $0 \notin \Delta^0_{4,2}$ and proposition 3.5 cannot be used. This information can be condensed in the two following short exact sequences

$$0 \longrightarrow \mathsf{I}^6_{4,2} \longrightarrow \mathsf{S}^4_{4,2} \longrightarrow \mathsf{I}^4_{4,2} \longrightarrow 0,$$

$$0 \longrightarrow \mathsf{I}^4_{4,2} \longrightarrow \mathsf{S}^0_{4,2} \longrightarrow 0.$$

The computation of the matrix $\mathbf{D}$ is now straightfoward. The (ordered) sets $\Delta_{4,2} = \{0,2,4,6\}$ and $\Delta^0_{4,2} = \{2,4,6\}$ index the rows and the columns respectively and $\mathbf{D}$ is thus $4 \times 3$. Since both $\mathsf{S}^2_{4,2}$ and $\mathsf{S}^6_{4,2}$ are irreducible, the lines $d = 2$ and $d = 6$ of $\mathbf{D}$ contain a single non-zero element: $\mathbf{D}_{2,2} = \mathbf{D}_{6,6} = 1$. From the first exact sequence above, the composition series is $0 \subset \mathsf{R}^4_{4,2} \subset \mathsf{S}^4_{4,2}$ with quotients $\mathsf{S}^4_{4,2}/\mathsf{R}^4_{4,2} \simeq \mathsf{I}^4_{4,2}$ and $\mathsf{R}^4_{4,2} \simeq \mathsf{I}^6_{4,2}$, in other words, $\mathbf{D}_{4,4} = \mathbf{D}_{4,6} = 1$. The second sequence provides $\mathbf{D}_{0,4} = 1$. Every other term is 0 and that completes the search for the composition matrix, which in turn gives the Cartan matrix via theorem 4.13:

$$\mathbf{D} = \begin{pmatrix} 0 & 1 & 0 \\ 1 & 0 & 0 \\ 0 & 1 & 1 \\ 0 & 0 & 1 \end{pmatrix}, \qquad \text{and} \qquad \mathbf{C} = \mathbf{D}^t\mathbf{D} = \begin{pmatrix} 1 & 0 & 0 \\ 0 & 2 & 1 \\ 0 & 1 & 2 \end{pmatrix}.$$

Lemma 4.12 gives, for the projective $P_{4,2}^6$, a filtration

$$0 \subset M_1 \subset M_2 \subset P_{4,2}^6,$$

with at most two intermediate modules $M_1$ and $M_2$, because **C** gives three composition factors: $I_{4,2}^4$ (once) and $I_{4,2}^6$ (twice). Since $P_{4,2}^6$ is the projective cover of $I_{4,2}^6$, the rightmost quotient $P_{4,2}^6/M_2$ must be a cellular module whose head is the irreducible $I_{4,2}^6$. There is only one choice possible and $P_{4,2}^6/M_2 \simeq S_{4,2}^6$. That leaves two composition factors: $I_{4,2}^6$ and $I_{4,2}^4$. The irreducible $I_{4,2}^6$ appears as composition factor only in $S_{4,2}^4$ and $S_{4,2}^6$. However the next quotient $M_2/M_1$ cannot be $S_{4,2}^6$ because the cellular module with $d = 6$ has already appeared and cannot appear again according to lemma 4.12. Moreover $M_2/M_1$ cannot be either $I_{4,2}^4$ which is not cellular. So $M_1$ must be zero and the quotient $M_2/M_1 \simeq S_{4,2}^4$. The filtration is thus $0 = M_1 \subset M_2 = S_{4,2}^4 \subset P_{4,2}^6$. (Note that this filtration, given by lemma 4.12, is not a composition series as $S_{4,2}^4$ is reducible. But $0 \subset R_{4,2}^4 \subset S_{4,2}^4 \subset P_{4,2}^6$ is such a composition series where indeed $R_{4,2}^4 \simeq I_{4,2}^6$, $S_{4,2}^4/R_{4,2}^4 = I_{4,2}^4$ and $P_{4,2}^6/S_{4,2}^4 \simeq I_{4,2}^6$.) The filtration $0 \subset S_{4,2}^4 \subset P_{4,2}^6$ indicates that the short sequence

$$0 \longrightarrow S_{4,2}^4 \longrightarrow P_{4,2}^6 \longrightarrow S_{4,2}^6 \longrightarrow 0$$

is exact, and it does not split since the projective cover of $I_{4,2}^6$ is indecomposable. The same reasoning for $P_{4,2}^4$ yields the filtration $0 \subset S_{4,2}^0 \subset P_{4,2}^4$ and the short non-split exact sequence:

$$0 \longrightarrow S_{4,2}^0 \longrightarrow P_{4,2}^4 \longrightarrow S_{4,2}^4 \longrightarrow 0.$$

These examples cover the main ideas of the proof of the following theorem.

**Proposition 4.14.** *The set $\{P_{n,k}^d \mid d \in \Delta_{n,k}^0\}$ forms a complete set of non-isomorphic indecomposable projective modules of $b_{n,k}$. When $d$ is critical or when there is no $d^-$ forming a symmetric pair with $d$, then $P_{n,k}^d \simeq S_{n,k}^d$; otherwise, $P_{n,k}^d$ satisfies the non-split short exact sequence*

$$0 \longrightarrow S_{n,k}^{d^-} \longrightarrow P_{n,k}^d \longrightarrow S_{n,k}^d \longrightarrow 0. \tag{4.11}$$

*Proof.* When $d$ is critical or $d$ is alone in its orbit $[d]$, $S_{n,k}^d$ is irreducible and appears as composition factor in no other cellular modules by proposition 4.10. Its line in the matrix **C** thus contains a single non-zero element and $P_{n,k}^d = S_{n,k}^d = I_{n,k}^d$.

Let $d$ be non-critical and suppose that $[d]$ contains at least one element distinct from $d$. Proposition 4.10 gives the non-zero composition multiplicities: $\mathbf{D}_{d,e}$ is non-zero and equal to 1 if and only if $e$ is either $d$ or $d^+$. (Of course $d^+$ must belong to $\Delta_{n,k}$ for $\mathbf{D}_{d,d^+}$ to be non-zero.) Theorem 4.13 gives

$$\mathbf{C}_{d,e} = [P_{n,k}^d : I_{n,k}^e] = \sum_{f=0}^{\min(d,e)} \mathbf{D}_{f,d}\mathbf{D}_{f,e}. \tag{4.12}$$

Suppose first that $d^-$ is not in $\Delta_{n,k}$. Then $\mathbf{D}_{f,d}$ is non-zero only for $f = d$ and

$$\mathbf{C}_{d,d} = \mathbf{D}_{d,d} \times \mathbf{D}_{d,d} = 1, \qquad \text{and} \qquad \mathbf{C}_{d,d^+} = \mathbf{D}_{d,d} \times \mathbf{D}_{d,d^+} = 1$$

and all other $\mathbf{C}_{d,f}$, $f \in \Delta_{n,k}^0$, are zero. The projective $P_{n,k}^d$ will thus have precisely two composition factors, $I_{n,k}^d$ and $I_{n,k}^{d^+}$, and $I_{n,k}^d$ is the head of $P_{n,k}^d$ since the latter is the projective cover of the former. One possibility for the filtration given in lemma 4.12 is $0 \subset I_{n,k}^{d^+} \subset P_{n,k}^d$, but the quotient $P_{n,k}^d/I_{n,k}^{d^+} \simeq I_{n,k}^d$ is not a cellular as required by the lemma. The only other possibility

is $0 \subset \mathsf{P}^d_{n,k}$ and the quotient $\mathsf{P}^d_{n,k}/0$ must be isomorphic to a cellular module, according again to lemma 4.12. It can be only $\mathsf{S}^d_{n,k}$ and $\mathsf{P}^d_{n,k} = \mathsf{S}^d_{n,k}$.

Suppose finally that $d^-$ belongs to $\Delta^0_{n,k}$. As $\mathbf{D}_{d,d} = 1$ and $\mathbf{D}_{d^-,d} = 1$, the sum (4.12) gives

$$\mathbf{C}_{d,d} = \mathbf{D}_{d^-,d} \times \mathbf{D}_{d^-,d} + \mathbf{D}_{d,d} \times \mathbf{D}_{d,d} = 2 \qquad \text{and} \qquad \mathbf{C}_{d,d^-} = \mathbf{D}_{d^-,d} \times \mathbf{D}_{d^-,d^-} = 1.$$

Moreover, if $d^+ \in \Delta_{n,k}$, then there will also be a contribution $\mathbf{C}_{d,d^+} = 1$ as in the previous case. This means that $\mathsf{P}^d_{n,k}$ has up to four composition factors : $\mathsf{I}^{d^-}_{n,k}$, $\mathsf{I}^d_{n,k}$ twice and, if $d^+ \in \Delta_{n,k}$, $\mathsf{I}^{d^+}_{n,k}$. The filtration $0 \subset M_1 \subset \cdots \subset M_{\delta-1} \subset M_\delta = \mathsf{P}^d_{n,k}$ of lemma 4.12 is needed to close the argument. As $\mathsf{P}^d_{n,k}$ is the projective cover of $\mathsf{I}^d_{n,k}$, it follows that the quotient $\mathsf{P}^d_{n,k}/M_{\delta-1}$ must be a sum of isomorphic copies of $\mathsf{S}^d_{n,k}$. The composition factors of $\mathsf{S}^d_{n,k}$ are $\mathsf{I}^d_{n,k}$ and, if $d^+ \in \Delta_{n,k}$, $\mathsf{I}^{d^+}_{n,k}$. If $d^+ \notin \Delta_{n,k}$, the quotient $\mathsf{P}^d_{n,k}/M_{\delta-1}$ cannot be a sum of two copies of $\mathsf{S}^d_{n,k}$ as the only composition factor left would be $\mathsf{I}^{d^-}_{n,k}$ which is not cellular. So this first quotient $\mathsf{P}^d_{n,k}/M_{\delta-1}$ contains precisely one copy of $\mathsf{S}^d_{n,k}$, leaving the composition factors $\mathsf{I}^{d^-}_{n,k}$ and $\mathsf{I}^d_{n,k}$ to be accounted in the next quotients. Neither is by itself a cellular module, so they must form the next quotient $M_{\delta-1}/M_{\delta-2}$ and this quotient must be $\mathsf{S}^{d^-}_{n,k}$. The filtration then reads $0 \subset \mathsf{S}^{d^-}_{n,k} \subset \mathsf{P}^d_{n,k}$. The exactnesss of sequence (4.11) is thus proved and, since $\mathsf{P}^d_{n,k}$ is indecomposable, it does not split. ∎

Propositions 4.1, 4.9 and 4.14 end the proof of theorem 2.5.

# 5 Concluding remarks

The main results of this paper are described in section 2.4 and will not be repeated here. Instead the present remarks are devoted to list the main steps used to reach the results. A list of these key steps might help in the study of other algebras obtained from a cellular algebra $\mathcal{A}$ by left- and right-multiplication by an idempotent $P$. Here are these steps.

(1) Assuming that $\mathcal{A}$ is cellular, the algebra $\mathcal{B} = P\mathcal{A}P$ will be too by proposition 3.6 from König's and Xi's original result if $P^* = P$. The easy construction of the cellular datum for $\mathcal{B}$ relies however on further hypothesis on $P$, namely that the non-zero elements of $P$ im $C_{\mathcal{A}}P$ form a basis of $\mathcal{B}$. In the case of $\mathsf{b}_{n,k}$, this property was not too difficult to verify because the idempotent was a sum of the identity and elements with less through lines.

(2) The explicit formula (2.4) for the determinant of the Gram matrix was crucial. So was also the recursive expression of lemma 3.8 for the bilinear form $\langle \cdot, \cdot \rangle^d_\mathcal{B}$. This recursive formula played a role at several steps: the computation of dimensions of radicals and irreducible modules, the existence of non-zero morphisms inherited from those defined by Graham and Lehrer for $\mathsf{TL}_n$ and, in a roundabout way, the identification of the structure of the cellular modules in proposition 4.10.

(3) The proof of proposition 3.11 on the cyclicity of the cellular modules in the new algebra was used to get the non-split condition on the exact sequences of proposition 4.10. It relied heavily on a diagrammatic construction. Having all cellular modules to be cyclic is a remarkable property to hold and indeed Geetha and Goodman introduced the notion of cyclic cellular algebras [27] to describe such cellular algebras. Most interesting cellular algebras are cyclic, for example: Temperley-Lieb algebras, Hecke algebras of type $A_{n-1}$, cyclotomic Hecke algebra and $q$-Schur algebras. But, is $\mathcal{B} = P\mathcal{A}P$ cyclic if $\mathcal{A}$ is and $P$ is one of its idempotents? Or are there further conditions needed on the commutative ring $R$ or on the idempotent $P$?

Of course applying the present method to other algebras of the form $P\mathcal{A}P$, in particular when $\mathcal{A}$ is not a Temperley-Lieb algebra, may run into other difficulties. But the above three steps appear to be the main stumbling blocks.

## Acknowledgements

We thank Alexi Morin-Duchesne and David Ridout for their interest in the project and useful comments, Eveliina Peltola, Nicolas Crampé and Loïc Poulain d'Andecy for explaining their recent results, and Pierre-Alexandre Mailhot for his help encoding the diagrams. We also wish to thank the referees for bringing to our attention Jacobsen's and Saleur's early work on what would become the seam algebra and for their careful reading of the manuscript. ALR holds scholarships from the Fonds de recherche Nature et technologies (Québec) and from EOS Research Project 30889451, and YSA a grant from the Natural Sciences and Engineering Research Council of Canada. This support is gratefully acknowledged.

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
