# Peer review of "The representation theory of seam algebras"

_SciPost Physics, doi:SciPost Phys. 8, 019 (2020)_

## Round 1 · Referee Report · Anonymous (Referee 1) · 2019-11-11

Strengths

1/ Clear and pedagogical ;

Weaknesses

1/ Technical and specific subject ;

Report

The authors study the representations of an algebra recently introduced and called seam algebras when q (the parameter of the algebra) is generic or root of unity. They recall the representation theory of the Temperley-Lieb algebra then they state the main theorem about the representation theory of the seam algebra which mimics the ones of the Temperley-Lieb algebra. The rest of the paper is devoted to prove this result and is based on the notion of cellular algebra.

The paper is well-written and very pedagogical and can be used as a reference to learn about cellular algebra. Even though the subject is very specific and technical, the result and the method are interesting and deserve to be published.

Requested changes

1- in equation (2.9), the object with superscript " (k) " is not defined. 2- page 9 three lines after (3.4) lambda must be changed to beta

---

## Round 1 · Referee Report · Anonymous (Referee 2) · 2019-11-12

Strengths

1-Provides an exhaustive account on the representation theory of the so-called boundary seam algebras, both in the generic and the root of unity cases.

Weaknesses

1-The discussion of some related previous results could be better discussed.

Report

This paper considers a class of cellular algebras which were introduced under the name of boundary seam algebras by Morin-Duchesne, Ridout and Rasmussen. The authors recall results on the related Temperley-Lieb algebra and go on to set up the representation theory of the boundary seam algebras (going beyond Morin-Duchesne et al), both in the generic and the root of unity cases. The results are clearly stated in theorem 2.5, in the form of a complete set of irreducible standard modules in the generic case, and exact short sequences in the root of unity cases.

The paper is well written, interesting and clearly publishable. However, it would be helpful to the readers if the authors could clarify its connections to some earlier literature. First, the defining relations (2.7) are reminiscent of the cabling construction described in arXiv:math-ph/0611078, in the first lines of section 5 and figure 16. Second, is the determinant of the Gram matrix given here in proposition 2.4 related to any one of the constructions given in section 6.2 of arXiv:0709.0812, and can the Gram matrix be formulated in similar graphical terms?

Requested changes

1-Please comment on the relations to the papers cited in the report.

---

## Round 2 · Referee Report · Anonymous (Referee 2) · 2020-1-7

Report

The author's comments fully answer my questions and I now recommend publication.

---

## Round 2 · Author Response

Dear editor,

We thank the referees for their careful reading of the manuscript. Please see below a list of changes in this new version.

Sincerely,
Alexis Langlois-Rémillard and Yvan Saint-Aubin

---

## Round 2 · List of Changes

1) Typos noted by referees (and two others) corrected: superscript "(k)", an old notation, removed in equation (2.9); $TL(\lambda)$ changed to $TL(\beta)$ after (3.4), and two small grammar mistakes.

2) Third paragraph of introduction extended to introduce Jacobsen's and Saleur's contribution. (Second paragraph also slightly modified.)

3) Citations added when the Wenzl-Jones projector is defined.

4) Paragraph added at the end of section 2.2 to make precise the relationship between blob and seam algebras.

5) Figure 3, a graphical representation of the Lemma 3.8 inserted between this lemma and proposition 3.9.

6) Thanks to the referees added to the Acknowledgements.

---

## Editorial Decision

published